# Deficiency of exopolysaccharides and O-antigen makes *Halomonas bluephagenesis* self-flocculating and amenable to electrotransformation

Tong Xu[1], Junyu Chen[1], Ruchira Mitra[1,2], Lin Lin[1,3], Zhengwei Xie[4], Guo-Qiang Chen [5✉], Hua Xiang [1,2✉] & Jing Han [1,2✉]

*Halomonas bluephagenesis*, a haloalkaliphilic bacterium and native polyhydroxybutyrate (PHB) producer, is a non-traditional bioproduction chassis for the next generation industrial biotechnology (NGIB). A single-sgRNA CRISPR/Cas9 genome editing tool is optimized using dual-sgRNA strategy to delete large DNA genomic fragments (>50 kb) with efficiency of 12.5% for *H. bluephagenesis*. The non-essential or redundant gene clusters of *H. bluephagenesis*, including those encoding flagella, exopolysaccharides (EPSs) and O-antigen, are sequentially deleted using this improved genome editing strategy. Totally, ~3% of the genome is reduced with its rapid growth and high PHB-production ability unaffected. The deletion of EPSs and O-antigen gene clusters shows two excellent properties from industrial perspective. Firstly, the EPSs and O-antigen deleted mutant rapidly self-flocculates and precipitates within 20 min without centrifugation. Secondly, DNA transformation into the mutant using electroporation becomes feasible compared to the wild-type *H. bluephagenesis*. The genome-reduced *H. bluephagenesis* mutant reduces energy and carbon source requirement to synthesize PHB comparable to its wild type. The *H. bluephagenesis* chassis with a reduced genome serves as an improved version of a NGIB chassis for productions of polyhydroxyalkanoates (PHA) or other chemicals.

[1] State Key Laboratory of Microbial Resources, Institute of Microbiology, Chinese Academy of Sciences, 100101 Beijing, People's Republic of China. [2] International College, University of Chinese Academy of Sciences, 100049 Beijing, People's Republic of China. [3] College of Life Science, University of Chinese Academy of Sciences, 100049 Beijing, People's Republic of China. [4] Peking University International Cancer Institute, Health Science Center, Peking University, Beijing, People's Republic of China. [5] Center for Synthetic and Systems Biology, School of Life Sciences, Tsinghua University, Beijing, People's Republic of China. ✉email: chengq@mail.tsinghua.edu.cn; xiangh@im.ac.cn; hanjing@im.ac.cn

Next generation industrial biotechnology (NGIB) is a sustainable approach that aims at cost-effective production of various chemicals by overcoming the economic and technological challenges of current industrial biotechnology[1]. NGIB employs robust microbial chassis possessing unique characteristics to synthesize bio-based products. Hence, the selection and engineering of promising microbial strains play a pivotal role in NGIB[2]. *Halomonas bluephagenesis*, a rapidly growing haloalkaliphilic bacterium, has been developed as a low-cost and high yield chassis for NGIB to produce polyhydroxybutyrate (PHB) under open unsterile and continuous condition in seawater[3]. Previous work has been conducted on metabolic engineering of *H. bluephagenesis* for production of poly(3-hydroxybutyrate) and its copolymers like poly(3-hydroxybutyrate-*co*-3-hydroxyvalerate), poly(3-hydroxybutyrate-*co*-4-hydroxybutyrate), as well as chemicals, such as ectoine, L-threonine, and 3-hydroxypropionate[4–8]. Moreover, cultivation of this halophile has also been successfully scaled up to the 5000-L pilot scale fermentor, which proved its feasibility as an industrial chassis for the production of various bioproducts[4,9].

In recent years, increasing research are focused on developing engineered *H. bluephagenesis* strains with a higher production efficacy *via* synthetic biology[6,7,10]. Minimizing the genome size *via* deletion of nonessential genes can reduce the metabolic burden and impart more stability to the host cell[11]. Genome reduction can generate high-performing microbial strains possessing industrially important physiological characteristics. Several genome-reduced strains of *Escherichia coli*, *Bacillus* sp., *Corynebacterium glutamicum*, and *Streptomyces* sp. have been developed to improve their metabolite production efficiency[12,13]. Efficient genetic operating systems and genome editing tools are indispensable to accelerate the genome reduction in *H. bluephagenesis*. CRISPR/Cas9-based genome editing tool have been developed in *H. bluephagenesis*[14]. However, it is less effective for editing large genomic sequences. Another requirement for designing high-performance engineered strain is transformation of large DNA fragments for heterologous metabolic pathway reconstruction. Till now, conjugation is the only available method for DNA transformation in *H. bluephagenesis*[2]. This method allows the introduction of only plasmid DNA and not linear DNA fragments into the recipient cell. Moreover, DNA carrying capacity of plasmids are limited by their origin of replication. Therefore, introduction of large DNA fragment in *H. bluephagenesis* is not feasible by conjugation-based DNA transfer system. In this context, electroporation is a simple method that effectively transfers large-sized linear DNA fragments into host cell. Taken together, it is necessary to optimize the existing CRISPR/Cas9-based genome editing tool and enrich genetic transformation system to achieve a highly efficient genome-reduced strain of *H. bluephagenesis* that may serve as an excellent platform strain for NGIB.

Bacterial flagellum is a complex motor organelle composed of several proteins[15]. Biosynthesis of flagella and its rotation are energy-intensive processes. In *E. coli*, flagellar biosynthesis accounts for almost 2% of the biosynthetic energy consumption by the cell[16]. Interestingly, deletion of flagellar operon in *Pseudomonas putida* KT2440 improved ATP/ADP ratio by 30% and NADPH/NADP$^+$ ratio by 20%[16]. In addition, deletion of 76 genes relevant to flagella and pili formation increased polyhydroxyalkanoates (PHA) content and its yield by 45.6 and 73.4%, respectively, in *P. putida* KT2440[17]. The flagellar gene cluster in *H. bluephagenesis* is located upstream of a diguanylate cyclase gene. It is a 54.4 kb long fragment and does not contain any essential genes. Thus, it was assumed to be a suitable target to test the genome editing tool for large DNA fragment deletion in *H. bluephagenesis*. Moreover, it is expected that deletion of the flagellar gene cluster would reduce energy requirement of the cell by preventing flagellar biosynthesis.

Exopolysaccharides (EPSs) are extracellular, high-molecular-weight polymers that are crucial for biofilm formation. EPSs play essential roles in host-pathogen interactions and are believed to protect cells from environmental stress such as extreme pH, antibiotic or desiccation[18,19]. EPSs also act as autoagglutinins, which mediate microbial self-flocculation[20]. EPSs synthesis increases requirement of carbon source in the cell because the carbon supply is redistributed between EPSs biosynthetic pathway and other physiological processes. Knocking out the EPSs biosynthesis cluster in *Haloferax mediterranei* improved PHA production by 20%[21]. *H. bluephagenesis* exhibited viscous liquefaction phenotype when cultured on agar plate. This implied that *H. bluephagenesis* may secrete certain amounts of EPSs. It is plausible that the deletion of EPSs biosynthetic pathway would save carbon consumption for other physiological processes in this strain. Thus, the gene clusters involved in EPSs synthesis were selected for simplifying the genome of *H. bluephagenesis*.

This study aims to improve the properties of *H. bluephagenesis* *via* deletion of redundant synthesis pathways and make it more suitable as a chassis for NGIB.

## Results

### Optimization of CRISPR/Cas9-based genome editing system for large DNA fragment deletion

In a previous study, a CRISPR-Cas9-based gene editing system for *H. bluephagenesis* has been constructed[14]. However, the editing efficiency for large DNA fragments is not high enough. Previously, a 2.3 kb long DNA fragment was deleted with an efficiency of 16.67%[14]. Thus, a more effective genome editing tool is urgently needed for editing multiple genes and large DNA fragments. Type II CRISPR-Cas system, including Cas9, only induce double-strand break (DSB) at target site. It leaves a long gap that affects homology-directed repair for large fragments editing (Fig. 1a). In order to improve editing efficiency for large fragments, we introduced a double sgRNAs strategy with 1 kb homologous arms (Fig. 1b). To test the deletion efficiency, a 54.4 kb flagellar gene cluster containing 57 CDS (Coding sequence) was selected as the target (Fig. 1c and Supplementary Data 1). A 20 kb fragment was first edited to compare the efficiency between single and double sgRNAs strategies. Then, 20 to 50 kb fragments were edited with double sgRNAs to evaluate the efficiency of our developed strategy. As shown in Fig. 1c, the deletion efficiency of double sgRNAs (53.1%) was better than that of single sgRNA (31.2%). Besides, double sgRNAs caused mutation at a rate of 93.7%, whereas single sgRNA led to only 34.3% for editing 20 kb fragment. This ensured a higher probability of obtaining mutants using double sgRNAs when more complex fragments were knocked out. As expected, when double sgRNAs were used to delete 30, 40, and 50 kb fragments, the corresponding editing efficiencies were 25, 25, and 12.5%, respectively. The 50 kb of flagellar gene cluster was knocked out *via* our optimized CRISPR/Cas9-based genome editing tool, though deletion efficiency decreased with the increase in DNA fragment length.

### Deletion of capsular polysaccharide synthesis clusters

The wild-type strain of *H. bluephagenesis* exhibited viscous liquefaction phenotype when cultured on LB60 plate (Fig. 2a). This implied that *H. bluephagenesis* synthesized and secreted large amounts of EPSs outside the cell. Genome-scale bioinformatics analysis of *H. bluephagenesis* predicted a ~90 kb long gene cluster related to polysaccharide synthesis and efflux, and a 16 kb putative chondroitin synthesis gene cluster. The relative position of gene clusters on the genome were shown in Fig. 2b. As the

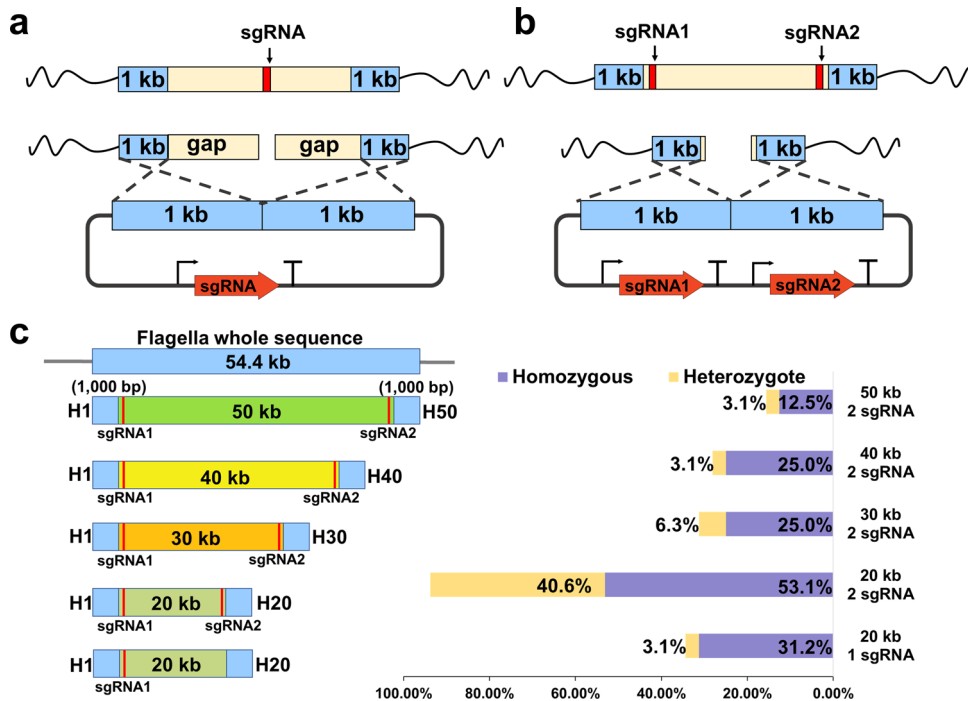

**Fig. 1 Flagellar gene cluster deletion using single or double sgRNAs. a, b** Design and comparison between single and double sgRNAs. The right-angle arrows and "T" symbols represent promoters and terminators, respectively. **c** Designed positions of homologous arms and sgRNAs for flagellar gene cluster deletion (left). Editing efficiency of deleting various flagellar gene cluster with single or double sgRNAs (right). 32 colonies in each group were verified by PCR. Homozygous genotype (purple) means the deletion of flagellar cluster in mutants which produce one PCR product with the size of 2.4 kb. Heterozygous genotype (yellow) represents the occurrence of both wild type and deletion of flagellar cluster in mutants which produce two PCR products with the size of 2.4 and 1.7 kb, respectively. The electrophoresis results of PCR products were shown in Supplementary Fig. 1. H1, the upstream homologous arm of flagellar gene cluster; H20 (H30, H40, and H50), the downstream homologous arm of flagellar gene cluster for the deletion of 20 kb (30, 40, and 50 kb) fragment.

presence of essential genes shown in Supplementary Data 2, the 90-kb long gene cluster was divided into four segments by five predicted essential genes (Fig. 2c and Supplementary Table 1). Firstly, two segments, named as PS1 (16,861 bp) and PS2 (22,861 bp), respectively, were chosen for knockout to interrupt extracellular polysaccharide synthesis. Unfortunately, deletion of individual PS1 (strain ΔPS1) or PS2 (strain ΔPS2), or both (strain ΔPS12), did not result in the loss of viscous liquefaction phenotype. Subsequently, the candidate gene cluster of chondroitin synthesis, named as PS3 (16,379 bp), was deleted. However, deletion of PS3 (strain ΔPS3) still had no obvious effect on the phenotypic change. Finally, the PS4 segment, which encoded four transporters, O-antigen ligase, and UDP-phosphate galactose phosphotransferase, was deleted based on ΔPS12. The resultant mutant ΔPS124 exhibited a dry phenotype when cultured on the LB60 plate (Fig. 2a). Interestingly, the liquid culture of ΔPS124 showed a rapid cell flocculation characteristic. With the settling time, more and more cells flocculated at the bottom of tube. Within 5 min, cell flocculation was visible. In 30 min, the cells settled down to the bottom leaving the liquid completely clear. In the case of the wild-type strain, culture was homogenous with no signs of flocculation (Fig. 2d).

**Effect of NaCl concentration on self-flocculation efficiency of ΔPS124.** The time curve of self-flocculation efficiency in 60MMG medium containing 1 M NaCl was shown in Fig. 3a. The self-flocculation efficiency achieved was 85% in 5 min, and it increased to ~96% in 15 min and attained stabilization. In contrast, flocculation efficiency was below 10% in TD1.0 strain. The sedimentation of the ΔPS124 cells due to self-flocculation without the aid of centrifugation would greatly facilitate the recovery of cells from culture after fermentation. Furthermore, as shown in Fig. 3b, self-flocculation efficiency of ΔPS124 was influenced by the NaCl concentration. Self-flocculation was obviously low (~10%) when NaCl concentration was below 0.2 M. With gradual increase in NaCl concentration to 0.6 M, self-flocculation efficiency increased to almost 90% and became stable with further increase in NaCl concentration to 1 M. In summary, ΔPS124 exhibited the best self-flocculation efficiency above 0.6 M NaCl.

**Exploring the self-flocculation mechanism of ΔPS124.** Bacterial autotransporters, exopolysaccharides, flagellin protein, Type IV pilus, or extracellular DNA are parts of biofilm. These components act as autoagglutinins, which result in self-flocculation[20]. However, in this study, knockout of EPSs clusters in *H. blue-phagenesis* resulted in self-flocculation of cells.

*Seeking possible agglutinins of ΔPS124.* Transcriptome analysis was carried out to provide some suggestions about the underlying mechanism of self-flocculation at molecular biology level. Self-flocculation is generally mediated by surface proteins, such as autotransporters or Type IV pilus[20]. Transcriptome analysis revealed that only 5 genes were significantly upregulated in the strain ΔPS124 (Supplementary Table 2). Among them, a filamentous hemagglutinin (FhaB), a 4-hydroxyphenylpyruvate dioxygenase (HppD), which was also annotated as hemolysin, and a hypothetical protein (Hyp) were initially speculated to be the key proteins related to self-flocculation. The remaining two upregulated genes encoded succinate-semialdehyde dehydrogenase (GabD) and gamma-aminobutyrate:alpha-ketoglutarate aminotransferase (DoeD), respectively. GabD, DoeD, and glutamate dehydrogenase (GdhA) participate in glutamate

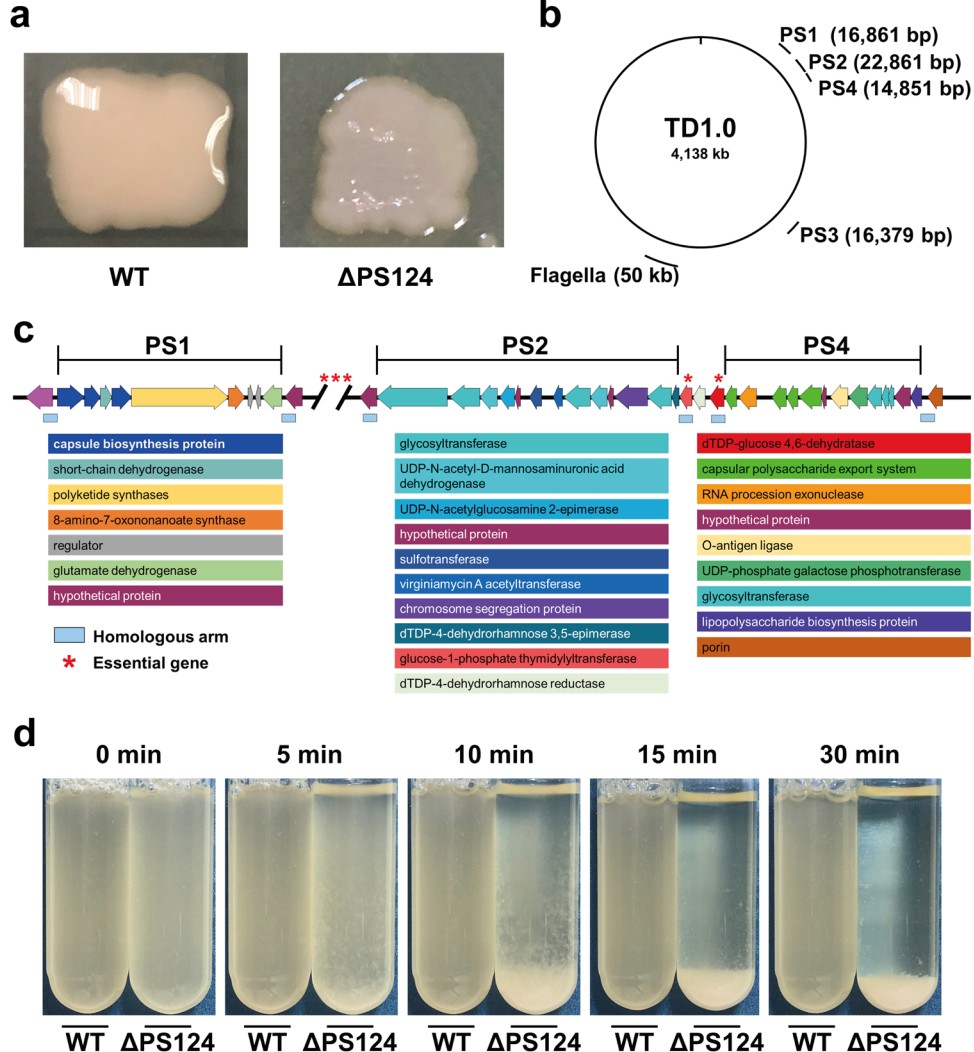

**Fig. 2 Deletion of EPSs synthesis pathway in *H. bluephagenesis*. a** Morphology of wild type and ΔPS124 cultured on LB60 agar plate. **b** The relative position of PS1, PS2, PS3, and PS4 and flagellar gene cluster in the genome of *H. bluephagenesis*. **c** The deleted EPSs clusters in ΔPS124. **d** Self-flocculation of ΔPS124 and TD1.0 after standing for 0–30 min. The strains were cultured in LB60 medium for 24 h and used for self- flocculation test. WT, *H. bluephagenesis* TD1.0; ΔPS124, *H. bluephagenesis* TD1.0 with deletion of PS1, PS2, and PS4 gene clusters; PS1, PS2, PS3, and PS4, the gene clusters involved in exopolysaccharide synthesis; Flagella, the gene cluster for flagella synthesis.

metabolism[22,23]. The individual deletion of *fhaB*, *hppD*, or *hyp* or simultaneous deletion of the 3 genes in ΔPS124 did not affect the self-flocculation (Supplementary Fig. 2).

Furthermore, transcriptome data showed that 81 genes were down-regulated in ΔPS124 (Supplementary Table 2). Most of them encoded membrane proteins. Among them, a nitrite reductase complex and a cytochrome d ubiquinol oxidase complex were significantly down-regulated. Both the enzymes are reported to function under anoxic conditions[24] or oxygen-limited conditions[25]. This suggests that the absence of EPSs may improve the oxygen permeability of cell membrane.

*Visible change in the cell surface of ΔPS124.* SEM analysis was subsequently conducted to observe the morphological differences between TD1.0 and ΔPS124. The cell surface of TD1.0 was rough and covered with wrinkles. On the contrary, the cell surface of ΔPS124 was smooth with some globular protuberances distributed on it (Fig. 4). The result indicated that the loss of EPSs exposed the cell surface of ΔPS124, which might be closely related with its self-flocculation. Simultaneously, TEM analysis was also performed to observe the changes in the outer membrane.

However, no obvious change could be observed (Supplementary Fig. 3a–d).

*Improved cell surface hydrophobicity of ΔPS124.* Enhanced cell surface hydrophobicity is one of the causes for self-flocculation[26]. The hydrophobicity of cell surface was measured by a modified phase partitioning assay. As expected, ΔPS124 had higher cell surface hydrophobicity (46.1% as measured by a modified phase partitioning assay) compared to TD1.0 (11.3%) in LB60 medium (Fig. 5). Notably, the hydrophobicity of ΔPS124 increased to 81.3% in 1.0 M NaCl solution, whereas the hydrophobicity of TD1.0 decreased to minus 47.8%. Similar results have been reported in *Halomonas elongata*[27], but the reason is not addressed and needs to be further clarified.

*Deficiency of both O-antigen and EPSs is the key factor for self-flocculation.* Self-flocculation was triggered when the PS4 cluster was knocked out in *H. bluephagenesis*, indicating that PS4 was closely related with self-flocculation. In order to clarify the specific genes directly attributing to self-flocculation, PS4 was divided into 3 parts (Part1, Part2, and Part3) to be cloned into

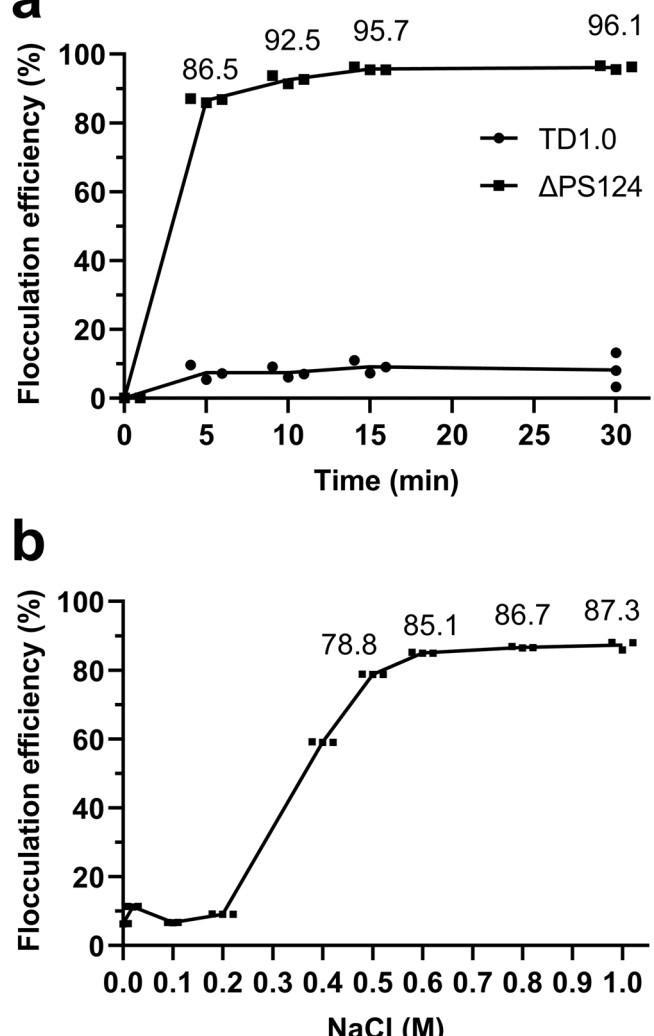

**Fig. 3 Flocculation efficiency of ΔPS124. a** Flocculation efficiency of TD1.0 and ΔPS124 in 60MMG medium containing 1 M NaCl. **b** Flocculation efficiency of ΔPS124 cultured in 60MMG medium and resuspended in different concentrations of NaCl solution (0, 0.02, 0.1, 0.2, 0.4, 0.5, 0.6, 0.8, 1.0 M). Data are shown from three technical replications. Error bars represent SD (*n* = 3). TD1.0, *H. bluephagenesis* TD1.0; ΔPS124, *H. bluephagenesis* TD1.0 with deletion of PS1, PS2, and PS4 gene clusters.

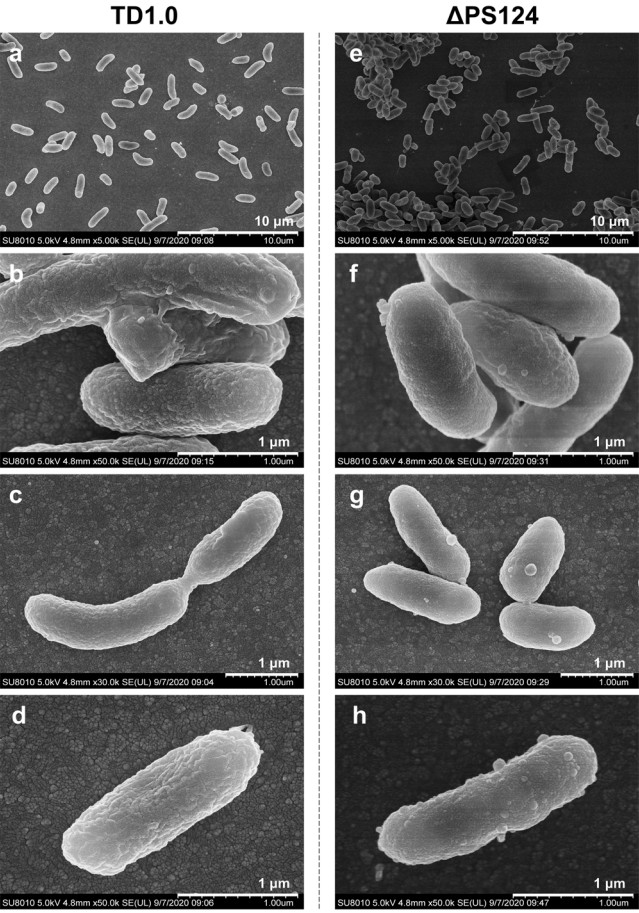

**Fig. 4 SEM images of TD1.0 and ΔPS124.** Strains TD1.0 (**a–d**) and ΔPS124 (**e–h**) were cultured in LB60 medium for 24 h before fixation. TD1.0, *H. bluephagenesis* TD1.0; ΔPS124, *H. bluephagenesis* TD1.0 with deletion of PS1, PS2 and PS4 gene clusters.

pSEVA321 vector (Fig. 6a). Several attempts to clone Part2 failed, which might be due to the toxicity of DUF4258 to the host cell (*E. coli*). The two constructed plasmids pSEVA321-Part1 and pSEVA321-Part3 were individually transformed into ΔPS124. The complementation of Part1 had no effect on self-flocculation (Supplementary Fig. 2c), whereas the complementation of Part 3 restored the full suspension of ΔPS124 (Fig. 6b). Simultaneously, the complementation of Part 3 maintained the cells with a dry phenotype when cultured on LB60 solid medium (Fig. 6b). Part3 encodes an O-antigen ligase (*waaL*), a UDP-phosphate galactose phosphotransferase (*rfbP*), a lipopolysaccharide biosynthesis protein (*wzz*), a hypothetical protein, and 3 glycosyltransferases (Fig. 6a). Among them, WaaL is the key enzyme for ligating O-antigen chains with the outer core of the lipopolysaccharides (LPS) and Wzz determines the length of O-antigen[28] (Fig. 6c). Both *waaL* and *wzz* serve as the boundary of Part3, implying that these 7 genes might constitute an O-antigen gene cluster. Subsequently, the O-antigen cluster was knockout in the wild type to test whether its absence is the only factor causing self-

flocculation. The mutant ΔO-antigen did not show a self-flocculation property in LB60 liquid medium and still exhibited viscous liquefaction phenotype on LB60 solid medium (Fig. 6b). Therefore, the presence of EPSs hindered self-flocculation even though O-antigen was deficient in *H. bluephagenesis*. These results demonstrated that self-flocculation of *H. bluephagenesis* could be attributed to the increase of cell surface hydrophobicity resulted from the simultaneous deficiency of EPSs and O-antigen.

**Deletion of PS3 gene cluster and 50 kb of flagellar cluster on the basis of ΔPS124.** In order to further reduce redundant gene clusters, the PS3 gene cluster (16,379 bp) was knocked out on the basis of ΔPS124, which resulted in a ΔPS1234 mutant. Subsequently, the 50 kb DNA fragment of flagellar cluster was knocked out on the basis of ΔPS1234, which resulted in a final ΔPS1234Δ50k mutant. In total, 121 kb DNA sequence was deleted *via* our optimized CRISPR/Cas9-based genome editing method, which accounted to ~3% of its whole genome. Interestingly, deletion of both PS3 gene cluster and 50 kb flagellar cluster did not affect the self-flocculation phenotype. This further confirmed that neither EPSs nor flagella acted as autoagglutinins and mediated the self-flocculation of our mutant strain.

The growth curves of TD1.0, ΔPS124 and ΔPS1234Δ50k cultured in LB60 medium were shown in Fig. 7a. The three strains had a similar growth pattern. However, the $OD_{600}$ of ΔPS124 and ΔPS1234Δ50k was lower than that of TD1.0 at stationary phase. The difference in OD values might be attributed to self-

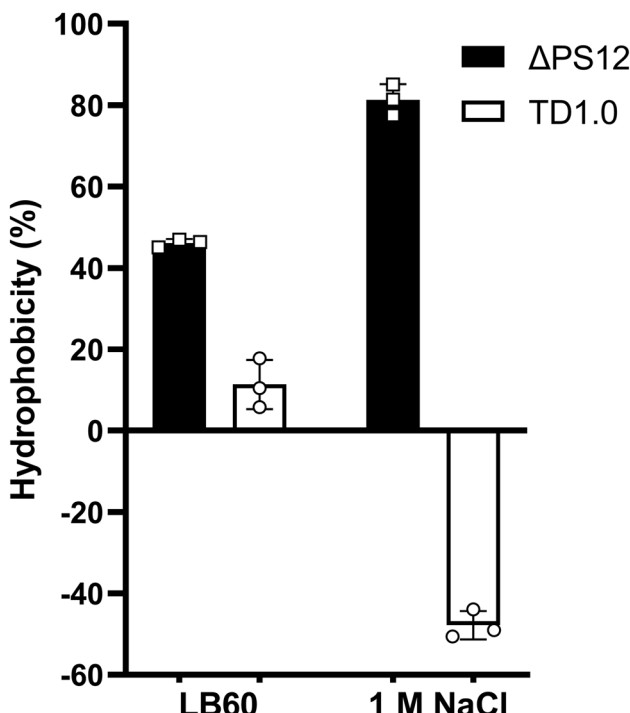

**Fig. 5 Cell surface hydrophobicity of ΔPS124 and TD1.0 in LB60 medium or 1 M NaCl solution.** LB60 medium contains 1 M NaCl. Data are shown from three biological replicates. Error bars are SD ($n = 3$). TD1.0, *H. bluephagenesis* TD1.0; ΔPS124, *H. bluephagenesis* TD1.0 with deletion of PS1, PS2 and PS4 gene clusters.

flocculation or the cell surface change (Fig. 4) of ΔPS124 or ΔPS1234Δ50k. Even though the culture was resuspended before measuring $OD_{600}$, the cells of ΔPS124 and ΔPS1234Δ50k rapidly flocculated and led to decreased OD values. Carbon source consumption and PHB synthesis of each strain were further analyzed at 36 h of fermentation in 60 MMG medium. The dry cell weight of the three strains were almost similar (~13 g/L). Likewise, the PHB production level of TD1.0, ΔPS124 and ΔPS1234Δ50k was also comparable (~9 g/L) (Fig. 7b). As shown in Fig. 7c, the total reducing sugar of ΔPS124 and ΔPS1234Δ50k in fermentation broth and supernatant was 1070 and 740 mg/L, and 700 and 600 mg/L, respectively. In contrast, the respective values of total reducing sugar for TD1.0 strain was 620 and 400 mg/L, respectively. This result indicated that mutant strain ΔPS124 could produce a similar amount of PHB by consuming less carbon source compared to TD1.0.

**ΔPS1234Δ50k transformed using electroporation.** Previously, conjugation was the only DNA transfer method for *H. bluephagenesis*[14]. Probably, EPSs secreted on the cell surface formed a barrier which limited the access of macromolecules. Therefore, it is conceivable that the absence of EPSs may promote the permeability of exogenous DNA into cells and thus increase the chances of electroporation-mediated gene transfer in *H. bluephagenesis*.

Considering the fact that *H. bluephagenesis* is a halophile and transformation medium with high osmolarity improved the electrotransformation efficiency of *Bacillus subtilis*[29], four kinds of media with different osmotic pressure (see "Materials and methods") were used to test the electrotransformation efficiency of *H. bluephagenesis* strains. Besides, two pulse voltages (1800 and

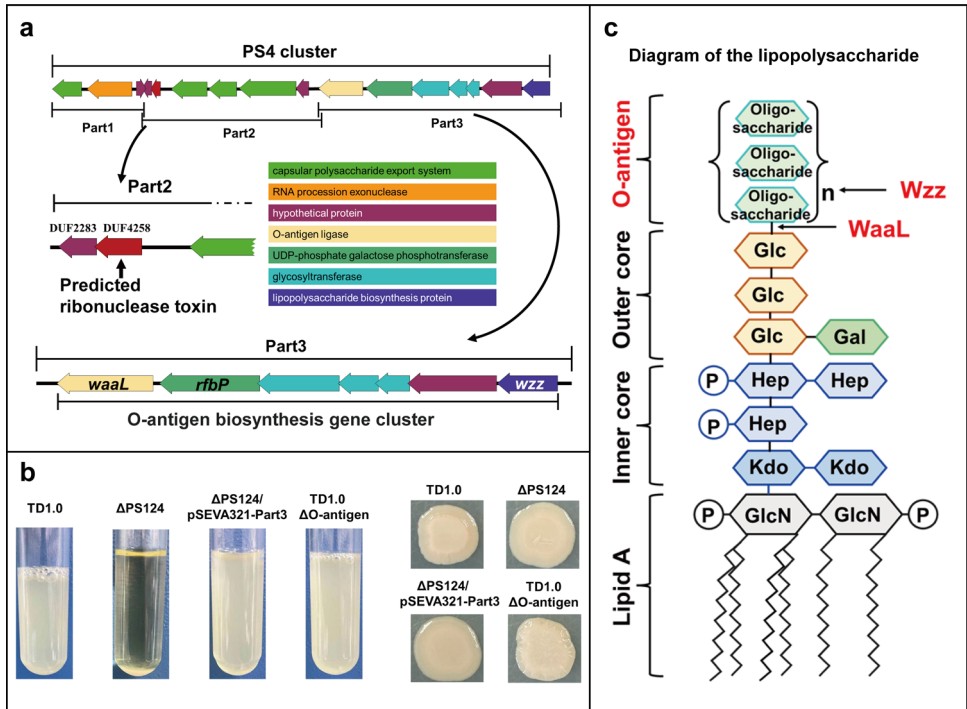

**Fig. 6 Deficiency of O-antigen is the key factor of self-flocculation. a** O-antigen biosynthesis genes in the PS4 cluster. **b** Self-flocculation of TD1.0, ΔPS124, ΔPS124/pSEVA321-Part3 and ΔO-antigen after standing for 30 min and their mucoid/dry phenotype. The strains were cultured in LB60 medium for 24 h and used for self-flocculation test (left). Morphology of TD1.0, ΔPS124, ΔPS124/pSEVA321-Part3, and ΔO-antigen cultured on LB60 agar plate (right). TD1.0, *H. bluephagenesis* TD1.0; ΔPS124, *H. bluephagenesis* TD1.0 with deletion of PS1, PS2, and PS4 gene clusters; ΔPS124/pSEVA321-Part3, ΔPS124 complemented with Part3 of the PS4 cluster; ΔO-antigen, TD1.0 with deletion of the 7 genes in Part3. **c** Diagram of the structure of lipopolysaccharide (LPS) in *E. coli*[28]. The action positions of WaaL and Wzz are depicted with arrows. GlcN, N-acetylglucosamine; Kdo, 3-deoxy-D-manno-octulosonate; Hep, heptose; Glc, glucose; Gal, galactose; Oligosaccharide, oligosaccharide unit of O-antigen; P, phosphate group.

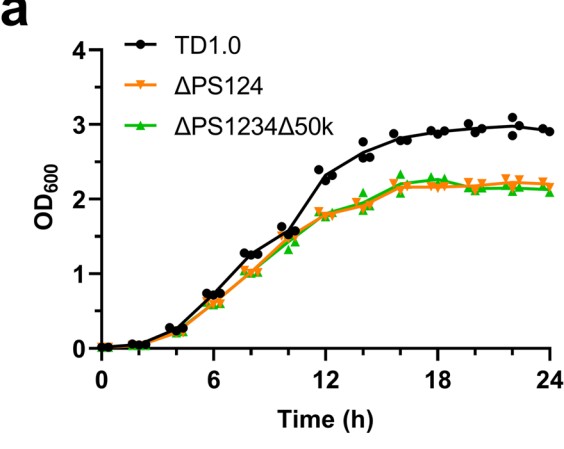

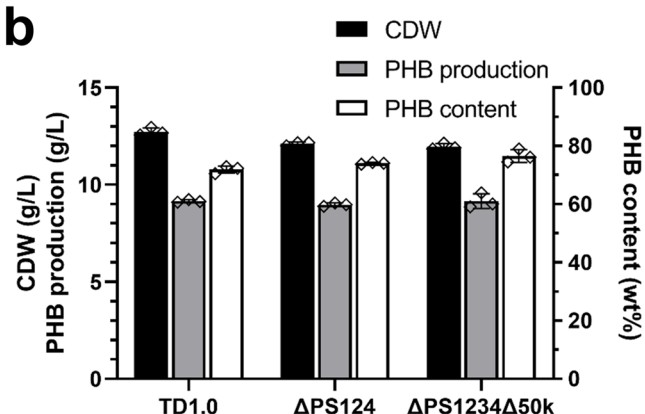

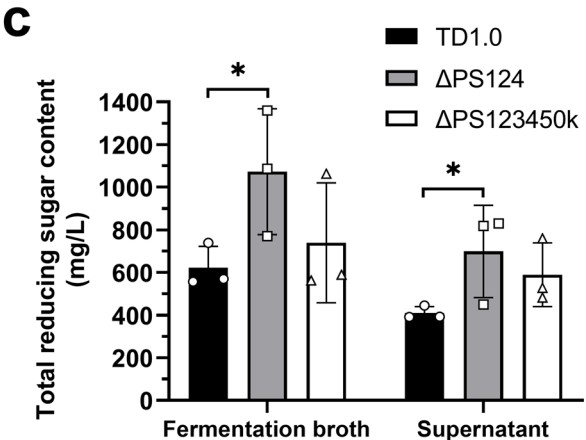

**Fig. 7 Comparison of growth and fermentation of TD1.0, ΔPS124 and ΔPS1234Δ50k. a** Growth curves. The strains were cultured in LB60 medium and growth was monitored at $OD_{600}$. **b** Shake-flask experiment of TD1.0, ΔPS124 and ΔPS1234Δ50k. Strains were cultured in 60 MMG medium at 37 °C for 36 h. **c** Total reducing sugar content after shake-flask experiment for 36 h. Data are shown from three biological replicates. Error bars are SD ($n = 3$ unpaired $t$ test, *$P < 0.05$). TD1.0, *H. bluephagenesis* TD1.0; ΔPS124, *H. bluephagenesis* TD1.0 with deletion of PS1, PS2, and PS4 gene clusters; ΔPS1234Δ50k, *H. bluephagenesis* TD1.0 with deletion of PS1, PS2, PS4, and PS3 gene clusters and flagellar gene cluster.

2500 V) were selected to test the best field strength. A total of 500 ng of plasmid pSEVA341 was transformed into the tested strains with a cell number of $1–2 \times 10^9$ by electrotransformation method. As shown in Fig. 8 and Table 1, the wild-type *H.*

*bluephagenesis* could not be transformed by electroporation under all the six tested electrotransformation conditions. In contrast, ΔPS1234Δ50k could be electrotransformed under five electrotransformation conditions. Low osmotic electroporation media with 0.5 M sucrose favored electroporation of ΔPS1234Δ50k. However, the transformation was inhibited in high or moderate osmotic medium comprising of 1 M sucrose or 1 M sorbitol and 0.5 M trehalose or 0.5 M sorbitol, 0.5 M mannose and 0.5 M trehalose. The highest pulse voltage of 2.5 kV showed a better transformation efficiency. Best electroporation efficiency of 400 colony-forming unit (cfu)/μg DNA was achieved by using 0.5 M sucrose at 2.5 kV.

In order to test if the plasmid was stable in the transformants, we extracted the plasmids from 50 transformants of ΔPS1234Δ50k. As shown in Fig. 8c and Supplementary Fig. 4, the plasmid pSEVA341 was transformed into ΔPS1234Δ50k by electrotransformation method and could replicate stably in this strain. This characteristic endows the possibility to further genetically engineer *H. bluephagenesis* and improve its performance as an excellent cell factory.

## Discussion

As summarized in Fig. 9, we optimized the CRISPR/Cas9 genome editing tool by using dual-sgRNA strategy for large-sized DNA fragment deletion in *H. bluephagenesis*. The gene clusters related to flagella, EPSs, and O-antigen biosynthesis in the strain were deleted using the optimized strategy, leading to a genome reduction by ~3%. Surprisingly, the resulting mutant showed two excellent characteristics. Firstly, the deletion of EPSs and O-antigen conferred self-flocculation characteristics to the strain without affecting growth and PHB production. Secondly, DNA transformation into the EPSs and O-antigen deficiency mutant could be mediated by electroporation, which otherwise was not feasible in the wild-type strain.

Minimizing the genome without affecting cell growth and unique robustness is still a challenge. Many species including *E. coli*, and *Bacillus* sp., have undergone an intensive genome reduction. In several cases, genome reduction has boosted the performance of the microbial strain by influencing their physiological traits and improving their metabolite producing ability. In *E. coli*, production of L-threonine in a genome-reduced strain was 83% higher than that in wild-type strain[5]. Likewise, a genome-reduced strain of *Bacillus amyloliquefaciens* lacking ~4.18% of the genome showed 10.4-fold increase in surfactin biosynthesis compared to the parent strain[30]. Interestingly, deletion of 7.7% of the genome of *Pseudomonas mendocina* NK-01 improved the ATP/ADP ratio of the cell by 11 times and enhanced MCL-PHA accumulation by 114.8%[31]. Based on these previous studies, genome reduction was attempted for the first time in *H. bluephagenesis* by using dual-sgRNA CRISPR/Cas9 genome editing tool. We knocked out the non-essential gene clusters associated with the production of EPSs and the formation of flagella in *H. bluephagenesis*. The genome of *H. bluephagenesis* was reduced by sequentially deleting flagella, EPSs, and O-antigen gene clusters, resulting into ~3% genome reduction and the obtainment of an improved chassis. Strikingly, the mutant strain gained the self-flocculation ability and electroporation feasibility, the latter of which is a beneficial characteristic in terms of industrial application.

NaCl concentration influenced the self-flocculation property of *H. bluephagenesis* ΔPS124. This can be explained by the Derjaguin, Landau, Verwey, and Overbeek (DLVO) theory, which is a classical theory of colloidal stability[32]. Bacteria usually exhibit a negative cell surface charge, which attracts the surrounding cations to form a double layer according to the DLVO theory[33].

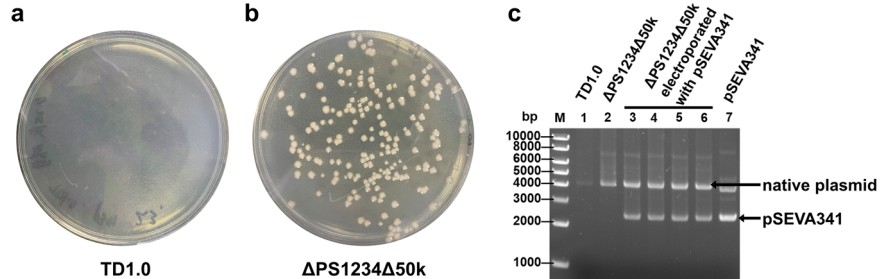

**Fig. 8 Electrotransformation of TD1.0 and ΔPS1234Δ50k. a, b** Cells were electrotransformed with 500 ng of pSEVA341. The LB60 agar plates (25 μg/mL of chloramphenicol) were incubated at 37 °C for 48 h. **c** Agarose gel electrophoresis of the plasmid extracted from TD1.0, ΔPS1234Δ50k, and electrotransformed ΔPS1234Δ50k. The black arrows show the bands of the native plasmid from *H. bluephagensis* and pSEVA341, respectively. TD1.0, *H. bluephagenesis* TD1.0; ΔPS1234Δ50k, *H. bluephagenesis* TD1.0 with deletion of PS1, PS2, PS4, and PS3 gene clusters and flagellar gene cluster.

**Table 1 Electrotransformation efficiency of TD1.0 and ΔPS1234Δ50k under different electroporation media and voltages.**

| Strain | Voltage (kV) | Transformation efficiency under different electroporation medium (cfu/μg DNA) | | | | |
| --- | --- | --- | --- | --- | --- | --- |
| | | 1.0 M sorbitol 0.5 M trehalose | 0.5 M sorbitol 0.5 M mannose 0.5 M trehalose | | 1.0 M sucrose | 0.5 M sucrose |
| TD1.0 | 2.5 | 0 | 0 | | 0 | 0 |
| | 1.8 | - | - | | 0 | 0 |
| ΔPS1234Δ50k | 2.5 | 2.7 ± 4.6 | 1.3 ± 1.2 | | 2.0 ± 2.0 | 320.7 ± 71.3 |
| | 1.8 | - | - | | 0 | 20.7 ± 1.2 |

-: The voltage was not considered. Three parallels were set for each group.

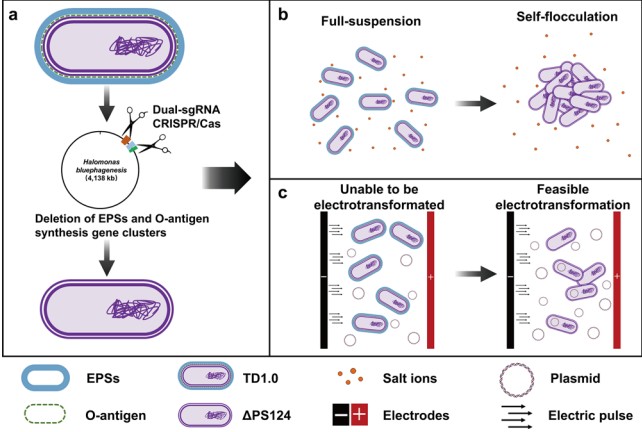

**Fig. 9 Illustration of the deletion of EPSs and O-antigen made *H. bluephagenesis* self-flocculating and electrotransformation feasible.**
**a** EPSs and O-antigen synthesis gene clusters of *H. bluephagenesis* TD1.0 were deleted with dual-sgRNA CRISPR/Cas strategy, and resulted in *H. bluephagenesis* ΔPS124. **b** *H. bluephagenesis* ΔPS124 can self-flocculate in the range of 0.4–1.0 M NaCl concentration, while *H. bluephagenesis* ΔPS124 cannot. **c** *H. bluephagenesis* ΔPS124 can be transformed by electroporation, whereas *H. bluephagenesis* ΔPS124 cannot.

The size of the double layer decreases with the increase of ionic strength. When the double layer decreases, the repulsion between cells also decreases, thus accelerating cell aggregation. When the ionic strength (NaCl concentration) was lower than 0.2 M, the self-flocculation of ΔPS124 was not observed. However, when the NaCl concentration increased to 0.6 M, self-flocculation led to faster sedimentation of cells as 90% of the cells settled down within 30 min (Fig. 3). Thus, it was possible to control the self-flocculation of the strain by adjusting salt concentration. Self-flocculation is very convenient for the recovery of fermentation products; therefore, it simplifies downstream processing during industrial production. Self-flocculating cells can be separated from the fermentation broth rapidly without centrifugation or membrane filtration. This saves the capital and operating costs and energy consumption. Moreover, after harvesting the cells, supernatant can be reused for repeated fermentation, and thus reducing generation of wastewater and effluent treatment cost[34]. In *Halomonas campaniensis* LS21, deletion of the electron transferring flavoprotein operon led to self-flocculation of the cells[34]. The self-flocculating *H. campaniensis* LS21 was cultured in a wastewaterless, open, unsterile, and continuous fermentation system for four runs for PHA production. However, the PHA production in the mutant was lower compared to its wild type[34]. In our case, the growth and PHB production of self-flocculating mutants of *H. bluephagenesis* (ΔPS124 and ΔPS1234Δ50k) were not impaired, although the loss of EPSs can be seen directly. Without synthesis of EPSs, cells may have saved a lot of carbon sources. Consistently, the total reducing sugar concentration in the fermentation broth for ΔPS124 and ΔPS1234Δ50k was higher than that of the wild type. Therefore, *H. bluephagenesis* ΔPS124 and ΔPS1234Δ50k could act as candidate chassis for the production of PHA and even other valuable bio-products.

Cell surface proteins acting as agglutinins are one of the causes for self-flocculation[35]. We tried to clarify the self-flocculation mechanism of *H. bluephagenesis* ΔPS124 by exploring differentially expressed genes which encode surface proteins by transcriptome analysis. Only 5 genes were upregulated significantly in ΔPS124. Among them, three genes possibly encoded surface proteins (adhesin, hemolysin, and one hypothetical protein). But individual or simultaneous deletion of these three genes in ΔPS124 could not influence its self-flocculation, implying they are not related with self-flocculation of ΔPS124. Since only two biological replicates were set in our transcriptomic analysis, it was possible to reduce the capacity for detecting differentially expressed genes (DEGs) between ΔPS124 and TD1.0. Our transcriptomic analysis found 30 DEGs with fold change greater than 2 and *P* value larger than 0.05. However, their reads numbers were all less than 140, lower than the average (890) or first quartile reads number (170) of our RNA-seq data. Thus, the 30

genes were not included as our targets for further genetic verification due to their low transcription level. Another possible autoagglutinin reported to cause self-flocculation is flagellar protein[36]. However, deletion of flagella gene cluster in *H. bluephagenesis* ΔPS124 had no visible effect on self-flocculation, indicating flagella is not the contributing factor to self-flocculation of ΔPS124. The lack of O-antigen has been reported to enhance the cell surface hydrophobicity of gram-negative bacteria[37]. Similarly, we found that the self-flocculation ΔPS124 showed increased surface hydrophobicity due to the O-antigen deficiency. An increase in cell surface hydrophobicity is a physical factor for self-flocculation. However, the ΔO-antigen mutant did not have the self-flocculation ability when the synthesis of EPSs was not disrupted. Our result indicates that EPSs might function as a shield to protect cells from self-flocculation of *H. bluephagenesis*. Similar function of capsular polysaccharides has been reported that it can block the function of the self-recognizing protein antigen[38], which is the key protein causing self-flocculation in *E. coli*, through physical shielding[38]. EPSs are usually one of the causes of self-flocculation[20,39,40], which is not the case for *H. bluephagenesis*. The disruption of O-antigen synthesis could lead to self-flocculation. The wzm ABC O-antigen transporter gene knockout mutant of *Synechococcus elongatus* was found to be capable of self-flocculation[41]. Collectively, the deficiency of EPSs and O-antigen is indispensable for self-flocculation of ΔPS124.

It is speculated that deletion of surface EPSs and O-antigen may enhance the permeability of the cell membrane and facilitate the movement of some substances into and out of the cells. So, we employed the NPN (1-N-phenylnaphthylamine) method to test the outer membrane permeability of our mutant strains. This method was used in many studies for outer membrane modification[42,43]. However, no difference was observed between wild type and ΔPS1234Δ50k (Supplementary Fig. 3e). This indicated that NPN might not be excluded by the EPSs or O-antigen of *H. bluephagenesis*. We further tested whether macromolecular plasmid DNA could be transformed into ΔPS1234Δ50k by electroporation. EPSs and O-antigen may be one of the reasons that prevented DNA from entering cells. Surprisingly, ΔPS1234Δ50k could be transformed using electroporation method. The electrotransformation method allows the introduction of linear DNA fragments and is urgently needed for high-throughput gene editing and mutant library screening in *H. bluephagenesis*. Using linear DNA fragments for transformation will save time and cost of plasmid construction. Therefore, cells that can be electrotransformed are the basis for large-scale genetic manipulation in the future. There are various methods of improving electrotransformation efficacy, ranging from physical parameter optimization to genetic manipulation. In *Dietzia* sp., optimization of the physical parameters including electric field strength, electroporation time, and chemical concentration obviously improved electrotransformation efficacy[44]. In addition, a thinner cell wall and deletion of the genes involved in peptidoglycan synthesis increased electrotransformation efficacy of *Corynebacterium glutamicum*[45,46]. Deletion of EPSs and O-antigen gene clusters realizing electrotransformation of *H. bluephagenesis* is a first report and a further modification to obtain a high-efficient electrotransformation strain is making progress.

In conclusion, CRISPR/Cas9 genome editing tool has been optimized by using dual-sgRNA to delete the redundant synthesis pathways of flagella, EPSs and O-antigen in *H. bluephagenesis*, and a self-flocculating mutant with less carbon and energy consumption was obtained. The mutant was able be accumulate similar amounts of PHB compared to its wild type. Notably, it is a breakthrough that deletion of EPSs and O-antigen gene clusters

realizes electrotransformation of *H. bluephagenesis*. Taken together, our work provides a non-traditional bioproduction chassis with great potential for NGIB to overcome economic and technological challenges of current industrial biotechnology.

## Methods

**Strains, plasmids, and culture conditions**. All strains and plasmids used in this study are shown in Supplementary Table 3. *H. bluephagenesis* TD1.0 was selected as the original chassis strain in this study[47]. *Escherichia coli* DH5α and S17-1 were used for plasmids construction and conjugation. *H. bluephagenesis* TD1.0 and its derived strains were cultured in LB60 medium (10 g/L tryptone, 5 g/L yeast extract, and 60 g/L NaCl)[8]. *E. coli* was cultured in LB medium (10 g/L tryptone, 5 g/L yeast extract, and 10 g/L NaCl). The compositions of the 60MMG medium are: 60 g/L NaCl, 30 g/L glucose, 1 g/L yeast extract, 2 g/L NH₄Cl, 0.2 g/L MgSO₄, 9.65 g/L Na₂HPO·12H₂O, 1.5 g/L KH₂PO₄, 10 ml/L trace element solution I and 1 ml/L trace element solution II. The compositions of trace element solution I are: 5 g/L Fe(III)–NH₄–citrate, 2 g/L CaCl₂, 1 M HCl. The compositions of trace element solution II are: 100 mg/L ZnSO₄·7H₂O, 30 mg/L MnCl₂·4H₂O, 300 mg/L H₃BO₃, 200 mg/L CoCl₂·6H₂O, 10 mg/L CuSO₄·5H₂O, 20 mg/L NiCl₂·6H₂O, 30 mg/L NaMoO₄·2H₂O. The pH of 60 MMG was adjusted to approximately 8.5 using 5 M NaOH[8]. The final concentration of 25 μg/ mL kanamycin, 25 μg/ mL chloramphenicol, or 100 μg/ mL spectinomycin was added as needed.

**Construction of plasmids**. Plasmids carrying CRISPR array and homologous arms (1000 bp each) were constructed by Gibson Assembly. The plasmid backbone was amplified using the primers pSEVA241F and pSEVA241R from pSEVA241[14]. Homologous arms were cloned from the genomic DNA of *H. bluephagenesis* by PCR. For the design of sgRNA spacer, a 20 bp of protospacer with ~50% GC content before NGG (PAM) was manually chosen using Snapgene. The CRISPR array for expressing sgRNAs were synthesized by GENEWIZ (Suzhou, China). The diagram for the construction of plasmids carrying CRISPR arrays and homologous arms is shown in Supplementary Fig. 5. To clone 3 parts of PS4, each part was amplified by PCR and then recombined into pSEVA321 by Gibson Assembly. The sequences of the primers and synthetic DNA are listed in Supplementary Table 4 and Supplementary Table 5.

**Conjugation**. *E. coli* S17-1 containing the target plasmid as the donor strain was cultured in LB medium supplemented with appropriate antibiotic. *H. bluephagenesis*, the recipient strain, was cultured in LB60 medium supplemented with appropriate antibiotics. The overnight cultures of both the strains were inoculated (1% v/v) into fresh LB and LB60 medium supplemented with appropriate antibiotics for 6 h, respectively. A total of 1 mL of each culture was harvested by centrifugation (6000 × *g*, 2 min), washed, and resuspended in 50 μL of LB or LB60 medium. The resuspended cells were mixed and dripped on LB20 agar plate. After overnight incubation at 37 °C, the moss was scraped up and resuspended in 100 μL of LB60 medium. Finally, the resuspended cells were spread on LB60 plate supplemented with appropriate antibiotics and incubated at 37 °C for 24–96 h to obtain colonies.

**Genome editing**. Genome editing was carried out based on a double-plasmid system[14]. Firstly, plasmid pQ08, which was designed to express Cas9 protein, was transformed into *H. bluephagenesis* by conjugation as described in 4.3. The derived plasmids of pSEVA241 carrying homologous arms and CRISPR arrays were transformed into *H. bluephagenesis*/pQ08 by conjugation. Recombinant colonies were screened on LB60 medium with 25 μg/mL chloramphenicol and 100 μg/mL spectinomycin. The colonies were randomly selected and identified by colony PCR using the primers shown in Supplementary Table 4.

**Determination of self-flocculation efficiency**. *H. bluephagenesis* strains cultured in LB60 medium for 24 h were used directly to measure the time curve of self-flocculation efficiency. ΔPS124 grown in 60MMG medium for 48 h was harvested, and then resuspended in the same volume of different concentrations of NaCl solution ranging from 0 to 1.0 M to measure the effect of salt concentration on self-flocculation efficiency.

For measurements, 15 mL culture was put into a 15 mL-centrifuge tubes, shaken thoroughly and then allowed to stand 0–30 min. A total of 200 μL liquid was collected at 0.5 cm away from the liquid surface and then its OD₆₀₀ was tested by Synergy H4 Hybrid Multi-Mode Microplate Reader (BioTek, USA). For each experimental group, three parallel samples were set. The self-flocculation efficiency of each sample was calculated according to following equation:

$$\text{Flocculation efficiency}(\%) = (1 - b/a) \times 100\%$$

where *a* is the OD₆₀₀ before rest and *b* is the OD₆₀₀ after rest[34].

**Shake-flask culture of *H. Bluephagenesis***. Single colonies were cultured in LB60 medium overnight. Then the cultures were inoculated (1% v/v) into 50 mL fresh LB60 medium, and subcultured for 10 h to serve as seed cultures. The seed cultures

were inoculated (5% v/v) into 50 mL 60MMG medium containing 30 g/L glucose in a 500 mL shake flask. Shake flasks were incubated under 200 rpm at 37 °C for 48 h. 10 mL of cultures were harvested by centrifugation and washed once with 10 mL ddH$_2$O. The cell pellets were lyophilized for 36 h and then weighed to calculate cell dry weight (CDW). Shake-flask culture was conducted in triplicates.

**Asssays of total sugar concentration**. The total sugar concentration was determined using the anthrone reagent[48]. 1 mL of anthrone reagent was added into 0.5 mL of sample, and treated in boiling water for 10 min. Then, OD$_{620}$ was measured using a microplate reader. Calibration curves were constructed with 0–100 mg/L glucose.

**Scanning electron microscopy (SEM)**. For SEM, cells cultured in LB60 medium for 24 h were harvested by centrifugation at $5000 \times g$ for 3 min, and then washed with PBS buffer (pH 7.2). Subsequently, the cells were fixed with 1 mL of 2% glutaraldehyde dissolved in PBS buffer (pH 7.2) and kept at room temperature for 30 min and then at 4 °C overnight. Then, the cells were washed with PBS buffer three times. After that, fixed cells were dehydrated by using different gradients (v/v) of ethanol solutions, 50, 70, 80, 90, and 100%. Samples were vacuum dried and coated with Au before SEM visualization (SU8010, Hitachi, Japan).

**Hydrophobicity assay**. The hydrophobicity of cell surface was determined by a modified phase partitioning assay based on microbial adhesion to hydrocarbon (MATH)[26]. Tested strains were cultivated for 24 h in LB60 medium. The cultures were centrifuged and resuspended in 1 M NaCl solution. A total of 5 mL suspension or cell culture was dispensed into a 15 mL tube. After a brief shake, 1 mL of the cell suspension or culture was taken out to get the initial OD$_{600}$. Then, 1 mL of xylene was added into the residual sample in 15 mL tube. The tubes were vortexed thoroughly for 1 min. After a complete phase separation, the lower aqueous phase was taken to obtain the ultimate OD$_{600}$. The cell surface hydrophobicity was calculated as (initial OD$_{600}$-ultimate OD$_{600}$)/initial OD$_{600}$ × 100%[27]. For each group, three replicates were set.

**Transcriptome analysis**. *H. bluephagenesis* TD1.0 and ΔPS124 were cultured in 60MMG medium under 200 rpm at 37 °C for 24 h. Cells were harvested by centrifugation at $10,000 \times g$ for 2 min and stored in a refrigerator at −80 °C before sequencing. Each group has two replicates. RNA extraction, transcriptome sequencing, data analysis, and differential expression analysis were performed by Novogene Co., Ltd. (Beijing, China)[49]. Briefly, total RNA extraction was performed by using TRIzol reagent (TIANGEN Biotech. (Beijing) Co., Ltd.) according to the manufacturer's instructions. mRNA was purified from total RNA using poly-T oligo-attached magnetic beads. Sequencing was performed on an Illumina Novaseq platform and 2.5 Gb data was generated for each sample. A *P* value of 0.05 and | log$_2$(fold change) | of 1 were set as the thresholds for significantly differential expression. Differential expression analysis between TD1.0 and ΔPS124 was performed using the DESeq R package (1.18.0). The RNA-seq data was deposited in the National Center for Biotechnology Information (NCBI) database under accession number PRJNA721089.

**PHB content analysis via gas chromatography**. A total of 30–40 mg lyophilized cells was collected in an esterification tube. Then, 2 mL esterification solution (97% methanol, 3% H$_2$SO$_4$, 1 g/L benzoic acid) and 2 mL chloroform were mixed and added into the esterification tube. Esterification tubes were heated in a hot air oven at 100 °C for 4 h. After the tubes were cool down to room temperature, 1 mL ddH$_2$O was added in each tube. The samples then were thoroughly mixed and allowed to stand 1 h for phase separation. Then, the lower organic phase was analyzed by GC-6820 (Agilent, USA) equipped with a DB-FFAP column (30 m × 0.32 mm × 0.25 μm film thickness, part number 123-3232, Agilent, USA). High purity N$_2$ was used as carrier gas and the column head pressure of N$_2$ carrier gas was set to 10 psi. A total of 1 μL of sample was injected with splitless mode. The temperatures of flame-ionization detector and injection port were set at 220 °C and 200 °C, respectively. The temperature of the column was set at 80 °C and was maintained for 1.5 min. Then, it rose to 140 °C at 30 °C/min and further to 220 °C at 40 °C/min, and was maintained at 220 °C for 0.5 min.

**Electroporation studies of *H. bluephagenesis***. *H. bluephagenesis* TD1.0 and ΔPS1234Δ50k were cultured overnight and inoculated (2% v/v) into 100 mL of fresh LB60 medium. The cultures were grown at 37 °C with shaking at 200 rpm for 5 h. Cells were harvested by centrifugation at $5,000 \times g$ for 10 min and then washed three times with electroporation medium. Four kinds of electroporation media with different osmolarity were used (0.5 M sucrose with lower osmolarity; 1.0 M sucrose solution with moderate osmolarity; a mixture of 1.0 M sorbitol and 0.5 M trehalose with high osmolarity; a mixture of 0.5 M sorbitol, 0.5 M mannose and 0.5 M trehalose with high osmolarity). Each 50 mL of original culture was suspended in 2 mL of electroporation medium. For electroporation, 200 μL of the competent cells ($1-2 \times 10^9$ cells) were mixed with 5 μL of pSEVA341 plasmid (100 ng/μL) and then transferred to an electroporation cuvette (0.2 cm gap, Bio-Rad). After incubation for 5 min, the cuvettes were exposed to a single electrical pulse using a Gene-Pulser

II (Bio-Rad) set at 25 μF, and 200 Ω. The above operations were carried out at 4 °C. Immediately following the electroporation, 1 mL of SOC60 recovery medium (20 mM glucose, 20 g/L tryptone, 5 g/L yeast extract, 60 g/L NaCl, 2.5 mM KCl, 5 mM MgCl$_2$, and 5 mM MgSO$_4$, pH 7.5) was added into the cuvette, mixed, and transferred to a 1.5 mL centrifuge tube. After incubation at 37 °C and 200 rpm for 4–5 h, the cells were plated on LB60-agar plates containing 25 μg/mL chloramphenicol for selection of transformed cells. Finally, the plates were incubated at 37 °C for 48 h, and the transformants were enumerated. A total of 50 transformants were randomly picked up and cultured overnight in LB60 liquid medium supplemented with chloramphenicol (25 μg/mL), and then the plasmids were extracted with Plasmid Extraction Kit and verified by agarose gel electrophoresis.

**Statistics and reproducibility**. Data was analyzed with GraphPad Prism software and shown as means ± standard deviation (SD). Statistical significance was performed using unpaired $t$ test at three significance levels ($*P < 0.05$).

**Reporting summary**. Further information on experimental design is available in the Nature Research Reporting Summary linked to this paper.

## Data availability

The RNA-seq data was deposited in the National Center for Biotechnology Information (NCBI) database under accession number PRJNA721089. All other data are available from the corresponding author on reasonable request. Source data underlying all figures have been provided as Supplementary Data 3.

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

## Acknowledgements

We are grateful to Prof Victor de Lorenzo for kindly donating the SEVA series plasmids. This study was supported by the National Key Research and Development Program (No. 2018YFA0900200), the National Natural Science Foundation of China (No. 31970031 and No. 32170031), and the China Postdoctoral Science Foundation (No. 2020M670496).

## Author contributions

T.X.: methodology, investigation, formal analysis, data curation, and writing - original draft. J.C.: Data curation and writing - original draft. R.M.: Data curation and writing - review & editing. L.L. and Z.X.: Methodology. G.-Q.C.: Resources and writing-review & editing. H.X. and J.H.: Methodology, supervision, and writing- review & editing.

## Competing interests

The authors declare no competing interests.
