## [Peer Review File · Communications Biology]

Reviewers' comments:

Reviewer #1 (Remarks to the Author):

The manuscript aims to present *H. bluephagenesis* as most promising cell factory for the next generation industrial biotechnology (NGIB) by deleting >50 kb of its gene cluster that codes for EPS biosynthesis genes using CRISPR-Cas and show its effect on self-flocculation capacity of the bacterium. Also, the attempt towards increasing the transformation capability of the bacterium via electroporation is an interesting read. However, authors have failed to provide convincing reasons for the reasons that led to increased flocculation of cells in response to EPS gene clusters deletion. Neither the transcriptomics data gives any indications.

Strengths:

1. The Abstract and sections of this manuscript are well-written, provide right motivation and clear explanations of the content. Clearly most relevant literature in the area has been surveyed.
2. Results has been presented in a logical manner
3. Though process supporting the manuscript is innovative and have applications to scientific community.

Weaknesses

4. Materials and methods section could have been improved. The authors have listed exhaustive tables in supplementary files, but without diagrams to explain the procedure, it is very difficult to follow the methodology.
5. Results section lacks proper explanations to the figures. It is very difficult to follow meanings and figures interpretations. Please consider adding abbreviations used in the figures, in the figures captions itself.
6. Discussions are weak. The authors have not given convincing reasons for the trends observed.

Questions to answer:

Q1: What is the concept/parameters authors have used to declare a gene essential or non-essential?

Q2: Do they have a list of genes designated as essential/non-essential in *H. bluephagenesis*?

Q3: EPS normally act as auto agglutinins which mediate microbial self-flocculation. In this manuscript, by deleting the EPS cluster, authors have shown increased self-flocculation. How? Any reasons? The discussion is not strong to make this observation or the science behind clear

Q4: What is a homozygous or heterozygous phenotype in Figure 1? No explanations have been given. Please add details related to these two terms in the manuscript.

Q5: Supplementary Figure 1c not all clear. What is meant by mutant band or wild type band? What does mutant band 2400 bp depict, etc.? Please elaborate and include more explanations in the text.

Q6: Figure 1B: what are the essential genes here? How was the lengths of clusters PS1, PS2, PS3, and PS4 decides? How has the authors decided, which genes should be chosen within the cluster (as non-essential), and which genes should be left out as essential (between the clusters)?

Q7: Manuscript mentions PS1, PS2, PS3, and PS4 to be involved in EPS synthesis. Please provide detailed table to the list of genes in this cluster, that the authors considered as to be involved in EPS synthesis. In short, there should be list of genes elaborating on the total EPS machinery this bacterium has. Example: How many total glycosyltransferases this bacterium has, and how is this class of genes distributed amongst the four clusters recognized?

Q8: Any reason for NaCl induced flocculation? Good to include few lines on the DLVO theory in the manuscript itself.

Q9: There is no discussions available for supplementary figure 2E. Please include few lines related to it in the manuscript

Q10: Deleting EPS should have increased cell density due to low energy requirements. However, manuscript mentions no effect on cell density of the muted vs the wild bacterium. Any thoughts on this? Include the reasons in the manuscript

Q11: What is the carbon source used in the experiments? Writing reducing sugars is misleading. Please include what are the sugars used in the experiments.

Q12: What is lane 1 and lane 2 in figure 6? What does the band at 4kb depict? It is seemed to be present in untransformed/non-electroporated cells too. Do the TD 1.0 strains have a native plasmid in

it?

Q13: Authors have mentioned in Line 312-increase in surface hydrophobicity leads to increased flocculation. But in this paper, authors have mentioned decrease in hydrophobicity of the muted cells (TEM images). This seems contradictory.

Q14: What is the novelty that authors claim in Line 337?

Q15: Include protocol for RNA extraction in section 4.9.

Q16: Include protocol for conjugation in Line 412.

Q17: Section 4.2 needs more work. A diagram is needed to explain

a. What regions of plasmid was amplified using the primers?

b. What was the arrangement of the CRISPR array, cas9, in the plasmid?

c. What are the promoters used?

d. There should be two different sgRNA's in the double cut experiments. How were these 2 sgRNA's located in relation to each other?

e. What are the constituents of the CRISPR array?

Please include the needful.

Q18: reference 13, line 391 have no reference to Vector F, and vector R. Please correct the reference or include more details in this manuscript.

Q19: Include information on how were the sgRNA sequences designed? What is the PAM sequence that is specific to this bacterium?

Q20: Line 201, is it figure 2E or Figure 3? Flocculation is depicted in Figure 3 supplementary.

Reviewer #2 (Remarks to the Author):

In the manuscript entitled "Deficiency of exopolysaccharides makes *Halomonas bluephagenesis* self-flocculating and electrotransformation feasible" the authors present the results of a study in which they adapted a CRISPR-Cas9 strategy to delete large sections of the *H. bluephagenesis* genome. Focus was placed specifically on removing genetic sections associated with the production of exopolysaccharides and the formation of flagella. These regions were selected because of the significant carbon and energetic demands associated with these compounds and organelles. The authors then went on to investigate the effects of these deletions on a range of characteristics, most notably flocculation and electroporation. The work expands from many previous studies from the authors, and the manuscript presents some valuable interesting findings on a non-traditional bioproduction chassis. The experiments are generally well designed (although some details are missing) however much of the analysis is quite speculative.

You can find below a list of the main concerns with the study.

1) There are numerous speculative statements throughout the manuscript, which in some cases weaken the quality of the analysis. Some examples include:

- lines 147-148: rephrase "because they might possibly be involved" as it is vague and seems unsubstantiated.

- lines 173-175: It is not clear that ionic strength is a key factor controlling flocculation. There seems to be a correlation but the experiments performed cannot validate the claim made. Also, the link to DLVO theory should be expanded on in the Discussion.

- lines 183-187: Again, the claim (that EPSs are the most critical factor in self-flocculation) is speculative and, while the evidence might be consistent with the claim, it does not prove it and

certainly cannot lead to such a strong statement. This argument is also somewhat discredited in lines 202-204, in which the authors speculate that the process is more complex than expected.

- lines 199-200: Speculative statement.

- lines 209-210: change to "This suggests that the absence of EPSs may improve the oxygen permeability of the cell membrane."

- lines 240-243: Could this be substantiated with experimental evidence?

- lines 319-323: Some speculation and a shortcut to the last statement.

- lines: 349-359: Some speculation and the last sentence could be considered subjective.

2) lines 61-64: "Minimizing the genome size via deletion of nonessential genes can reduce" and "Genome reduction can generate high-performing". Caution must be taken here; genome reduction does not systematically lead to such advantages. This is an important distinction and should be addressed in other parts of the manuscript as well (for example in the Discussion lines 276-280).

3) How extensive was the transcriptomic analysis? Much more information is required on how many replicates were used, how the samples were processed, how the differential expression analysis was performed, which platforms were used for analysis, etc. In the Materials and Methods, the authors divert these aspects by pointing to a third party. This is not sufficient.

4) The authors draw a direct relationship between a reduction in OD of the mutants compared to the TD1.0 strain (lines 223-230) and self-flocculation. However, there are no consideration that the smoother cell surface observed through microscopy (Figure 4) may lead to a reduction in light scattering, and hence a reduction in OD600 for a given cell number. This would also be consistent with the fact that similar dry cell weights were observed for all three strains even if OD differed.

Additional experiments investigating the relationship between OD and cell numbers for the mutants and TD1.0 would be required to confirm this and validate or reject the claim made by the authors.

5) It is unclear if statistical analysis was used when comparing values obtained between strains or culture conditions for various experiments. If so, more details should be provided; if not, it should be done. Examples include:

- lines 228-230: "The dry cell weight of the three strains were almost similar (~13 g/L)." This is also put in perspective in the following line where the authors state that TD1.0 had a "slightly higher biomass". Statistical analysis is required.

- lines 230-231: "Likewise, the PHA production level of TD1.0, Δ PS124 and Δ PS1234 Δ 50k was also comparable (~9 g/L)".

- lines 235-237: mutant strains "could produce a similar amount of PHA by consuming less carbon source". Again, without proper statistical comparisons of both PHA produced and carbon utilization the

claims and analyses made in the manuscript become highly speculative.

- other similar cases can be found throughout the manuscript.

6) line 124: the location of the gene cluster in the genome should be provided and the acronym CDS should be defined as it is encountered for the first time.

7) lines 129-130: It should be specified that the rate of mutation reported was specifically for deletions. Unless more information is provided, the authors have not shown whether this rate would also be observed when DNA fragments are inserted.

8) lines 249-250: How many cells were used? Should be included.

9) Table 1 should be reordered for clarity. It is quite confusing as is.

10) lines 260-265 + Figure 6: Why were only 4 transformants used for these experiments? This is quite low. And does this mean that the authors consider the transformation and stability to be 100%?

A larger number of transformants tested and more replication cycles would have allowed a better statistical assessment and, perhaps, more information on the long-term stability of the plasmid. There is no quantifiable information provided on these parameters.

11) Figure 3A: What was the ionic strength of the medium for the results shown? This should be included in the caption and in the corresponding section of the Discussion.

The same should be done for lines 343-344: The conditions used to test flocculation should be specified.

12) Were any experiments performed to see if the flocculation effects observed at different ionic strengths were linked to the growth of the cells or only to the given state of the medium (i.e. would cells grown in a medium at a given ionic flocculate at a different rate if they were centrifuged and resuspended in a new medium at a different ionic strength?)

13) lines 329-330: Individual deletion of the three genes did not affect self-flocculation, but the deletion of two or all three at once may have (which would highlight potential interactions).

14) line 341: Details of the NPN method should be included.

15) line 410: How was Cas9 transformed in the cells? Was conjugation used? Details should be included.

16) lines 419-420: include the range of NaCl concentrations used.

17) lines 428-429: Is there a reference for this equation?

18) lines 430: The title of Section 4.6 is wrong, it should refer to Cultures and/or Cell Dry Weight, not PHA production.

19) lines 465-466: General details of GC method used should be included.

20) lines 486-487: Were sequencing and PCR used?

21) The terms PHB, PHA and PHBV seem to be used interchangeably throughout the manuscript. This should be corrected to the proper molecule produced in the study.

22) The following points should also be addressed.

- line 24: the claim that *H. bluephagenesis* is "one of the most promising cell factories" is very subjective and unfounded. This should be toned down and rephrased here and at other locations in the manuscript. The same should be done at line 52 where "Extensive research" should be replaced by "Previous work" or "Previous studies have focused on".

- lines 28-29: "including those encoding flagella or exopolysaccharides (EPSs), were"

- line 38: "serves as an improved version"

- lines 59-60: Reference(s) needed.

- line 69: "tools"

- line 76: "their origin"

- lines 113-114: avoid the repetition of "editing"

- line 117: The use of "To our knowledge," is odd since at least one author was also an author on the article referenced.
- lines 118, 119, 122, 124, 177, 404, 406, 455, 470, 479, 480: some characters were replaced by empty checkboxes.
- line 164: Change the term "equilibrium". This is not a situation that leads to an equilibrium. It can lead to a stable state, but not equilibrium.
- line 173: "and lost the ability of self-flocculation below 0.2 M NaCl" as it is redundant with lines 168-170.
- throughout the manuscript: commas designating thousands in numbers should not be followed by a space (e.g. line 146: 16, 861 bp should be 16,861 bp).
- lines 198-199: Reference(s) needed.
- line 205: remove the comma.
- line 223: specify if the OD600 was final or at stationary phase.
- line 308: change "investment" to "capital and operating" costs.
- line 309: in some processes, the presence of inhibitors or by-products in the medium may limit or prohibit the reuse of medium/supernatant.
- line 313: remove comma.
- line 379: "are shown".
- lines 404-405: how was the moss blown up?

Reviewer #3 (Remarks to the Author):

This manuscript reports on the development of a strategy for an optimized CRISPR/Cas9 genome editing of a PHB-producing *H. bluephagenesis* strain. Specifically, the authors reported on the use of a dual-sgRNA to delete the redundant synthesis pathways of flagella and EPSs allowing the obtainment of a mutant with enhanced self-flocculation and less carbon and energy requirement for metabolic processes. The mutant was able to accumulate PHB at a percentage similar to WT strain and could be transformed by electroporation.

Overall, the paper is well written, the methodological approach is suitable and the discussion of the obtained results is appropriate. The findings of this work are very interesting considering the importance of next generation industrial biotechnology (NGIB) which aims at developing novel and sustainable approaches, through the selection and engineering of appropriate microbial strains, for the obtainment of chemicals/products overcoming economic and technological challenges of current industrial biotechnology.

I found the paper suitable for publication. I have however some specific comments:

1. Pg. 2, line 23 and Pg. 3, line 50: "bacterium" instead of "bacteria"
2. Pg. 4, lines 116, 118 and throughout the manuscript: please review the character used (example in the word "efficiency" there is a "square" instead of "ffi")

Point-to-Point Response to Reviewers' Comments

(Manuscript ID# COMMSBIO-21-1273A)

Reviewers' comments:

Reviewer #1 (Remarks to the Author):

The manuscript aims to present *H. bluephagenesis* as most promising cell factory for the next generation industrial biotechnology (NGIB) by deleting >50 kb of its gene cluster that codes for EPS biosynthesis genes using CRSIPR-Cas and show its effect on self-flocculation capacity of the bacterium. Also, the attempt towards increasing the transformation capability of the bacterium via electroporation is an interesting read. However, authors have failed to provide convincing reasons for the reasons that led to increased flocculation of cells in response to EPS gene clusters deletion. Neither the transcriptomics data gives any indications.

Strengths:

1. The Abstract and sections of this manuscript are well-written, provide right motivation and clear explanations of the content. Clearly most relevant literature in the area has been surveyed.
2. Results has been presented in a logical manner
3. Though process supporting the manuscript is innovative and have applications to scientific community.

Response: Great thanks for your positive comments. Based on your suggestions, we have performed additional experiments and demonstrated that the increase of cell surface hydrophobicity resulted from the simultaneous deficiency of EPSs and O-antigen is the reason that led to self-flocculation. The new results have been included in Line 190-192, Line 208-240, Fig. 5, Fig. 6 and Supplementary Fig. 2.

Weaknesses

4. Materials and methods section could have been improved. The authors have listed exhaustive tables in supplementary files, but without diagrams to explain the procedure, it is very difficult to follow the methodology.

Response: Thanks for your helpful comment. We have improved “Materials and methods” section by including a diagram as Supplementary Fig. 5 to explain how the psgRNA plasmid was constructed. Meanwhile, the detailed protocols for RNA extraction and gas chromatography have been added in “Materials and methods” section (Line 508-517, 527-531) to provide an easy-to-follow methodology.

5. Results section lacks proper explanations to the figures. It is very difficult to follow meanings and figures interpretations. Please consider adding abbreviations used in the figures, in the figures captions itself.

Response: Thanks for your comment. We have added the abbreviations used in the figures in their legends to make the meaning and figures clear in Line 720-728, 736-739, 746-748, 752-754, 782-785, 793-796.

6. Discussions are weak. The authors have not given convincing reasons for the trends observed.

Response: Thanks for your comments. We have done additional experiments and demonstrated that the increase of cell surface hydrophobicity resulted from the simultaneous deficiency of EPSs and O-antigen is the reason that led to self-flocculation. Meanwhile, based on your suggestion, we have improved the “Discussion” section by including more factors causing self-flocculation in Line 365-380.

Questions to answer:

Q1: What is the concept/parameters authors have used to declare a gene essential or non-essential?

Response: Theoretically, only the essentiality of enzymes but not regulators or transcription factors can be predicted through the genome scale metabolic model by Flux Balance Analysis. We have constructed a genome scale metabolic model from scratch (not published) and analyzed the enzymes that are forced to be greater than 0 to support cellular growth, and we termed these reactions to be essential. The single gene copy involved in such a reaction is assigned to be an essential gene.

Q2: Do they have a list of genes designated as essential/non-essential in *H. bluephagenesis*?

Response: A list of genes designated as essential in *H. bluephagenesis* has been provided as Supplementary data 2 for the review process.

Q3: EPS normally act as auto agglutinins which mediate microbial self-flocculation. In this manuscript, by deleting the EPS cluster, authors have shown increased self-flocculation. How? Any reasons? The discussion is not strong to make this observation or the science behind clear

Response: Thanks for your comment. As you mentioned, EPS normally act as auto agglutinins mediating self-flocculation. In our study, we found the knockout of 3 EPS clusters in *bluephagenesis* led to self-flocculation. Based on your comment, we have performed additional experiments to clarify the reason that led to self-flocculation of *H. bluephagenesis*. Self-flocculation was triggered by the knockout of PS4 cluster, indicating that PS4 is closely related with self-flocculation. So we divided the PS4 cluster into 3 parts and tested the effect of their complementation on the self-flocculation of Δ PS124. Our results revealed that the complementation of the O-antigen cluster in PS4 made the self-flocculation phenotype disappear. However, knockout of the O-antigen cluster in wild type did not trigger self-flocculation, implying the deficiency of both EPSs and O-antigen attributed to self-flocculation. We also clarified that the cell surface hydrophobicity of Δ PS124 was higher than that of the wild type. Thus, it is concluded that the lack of EPSs unshielded the more

hydrophobic cell surface resulted from O-antigen deficiency and consequently led to self-flocculation of Δ PS124. We have included the new data and discussion in Fig. 5-6 and Line 365-380 to improve the manuscript.

Q4: What is a homozygous or heterozygous phenotype in Figure 1? No explanations have been given. Please add details related to these two terms in the manuscript.

Response: Thanks for your comment. Homozygous genotype means the deletion of flagellar cluster in mutants which produce one PCR product with the size of 2.4 kb. Heterozygous genotype represents the occurrence of both wild type and deletion of flagellar cluster in mutants which produce two PCR products with the size of 2.4 kb and 1.7 kb, respectively. The detailed information has been added in the legend of Figure 1 to make it clear.

Q5: Supplementary Figure 1c not all clear. What is meant by mutant band or wild type band? What does mutant band 2400 bp depict, etc.? Please elaborate and include more explanations in the text.

Response: We have added more explanations in the legend of Supplementary Figure 1 as follows. The external primers were designed based on the upstream of upstream homologous arm and downstream of downstream homologous arm. The colonies with flagellar gene deletion (Mutant) could produce a 2.4-kb PCR product using the external primers. However, there was no PCR product for the wild-type strain using the external primers due to its theoretical size (>22 kb). The forward primer of inner primers was designed based on the upstream of the upstream homologous arm and the reverse primer on the flagellar gene cluster. The wild-type strain yielded a 1.7-kb PCR product using the inner primers. When only the band of 2.4 kb was detected, the colony was a homozygote. When both the bands of 1.7 kb and 2.4 kb were detected, the colony was a heterozygote.

Q6: Figure 1B: what are the essential genes here? How was the lengths of clusters PS1, PS2, PS3, and PS4 decided? How has the authors decided, which genes should be chosen within the cluster (as non-essential), and which genes should be left out as essential (between the clusters)?

Response: Thanks for your comments. The essential genes located at the space sequences between PS1, PS2, PS3 and PS4 and the list of genes in PS1, PS2, PS3 and PS4 are listed in Supplementary Table 1. The boundaries and the length of PS1, PS2, PS3 and PS4 were determined by the genes involved in the synthesis of exopolysaccharides and precursor of sugars. The speculation of essential genes refers to the response to Q1.

Q7: Manuscript mentions PS1, PS2, PS3, and PS4 to be involved in EPS synthesis. Please provide detailed table to the list of genes in this cluster, that the authors considered as to be involved in EPS synthesis. In short, there should be list of genes elaborating on the total EPS machinery this bacterium has. Example: How many total glycosyltransferases this bacterium has, and how is this class of genes distributed

amongst the four clusters recognized?

Response: Great thanks for your helpful suggestions. The genes in PS1, PS2, PS4 and PS3, which might be involved in exopolysaccharide synthesis, have been listed in Supplementary Table 1. *H. bluephagenesis* TD1.0 has a total of 31 genes encoding glycosyltransferases. Among them, five are distributed in PS2 and PS4. The exopolysaccharide synthesis protein encoding genes are distributed in the four PS clusters.

Q8: Any reason for NaCl induced flocculation? Good to include few lines on the DLVO theory in the manuscript itself.

Response: Thanks for your suggestion. We have added the reason for NaCl induced flocculation in “Discussion” section as follows (Line 325-332). NaCl concentration influenced the self-flocculation property of *H. bluephagenesis* Δ PS124. This can be explained by the Derjaguin, Landau, Verwey, and Overbeek (DLVO) theory, which is a classical theory of colloidal stability. Bacteria usually exhibit a negative cell surface charge, which attracts the surrounding cations to form a double layer according to the DLVO theory. The size of the double layer decreases with the increase of ionic strength. When the double layer decreases, the repulsion between cells also decreases, thus accelerating cell aggregation.

Q9: There is no discussions available for supplementary figure 2E. Please include few lines related to it in the manuscript

Response: Thanks for your suggestion. We observed no difference between wild type and Δ PS1234 Δ 50k. This indicated that NPN might not be excluded by the EPSs EPSs or O-antigen of *H. bluephagenesis*. Based on your comment, we have included this discussion in Line 386-387 and more details about NPN method in the supplementary method.

Q10: Deleting EPS should have increased cell density due to low energy requirements. However, manuscript mentions no effect on cell density of the muted vs the wild bacterium. Any thoughts on this? Include the reasons in the manuscript

Response: Thanks for sharing your thoughts. In our study, we did not observe increased cell density of EPS mutant. It might be due to the cell division speed of mutant did not increase. But the EPS mutant synthesized comparable PHB with the wild type by consuming less carbon source. This point was mentioned in Line 265-267.

Q11: What is the carbon source used in the experiments? Writing reducing sugars is misleading. Please include what are the sugars used in the experiments.

Response: Thanks for your correction. 30 g/L glucose was used as the carbon source in the experiments and the information has been added in Line 475. It is speculated that other reducing sugars might be converted from glucose and excreted. Thus, we tested the total reducing sugar including glucose in fermentation broth and supernatant to compare the glucose consumption by the mutants and wild type.

Q12: What is lane 1 and lane 2 in figure 6? What does the band at 4kb depict? It is seemed to be present in untransformed/non-electroporated cells too. Do the TD 1.0 strains have a native plasmid in it?

Response: Lane 1 and lane 2 represent the bands of plasmids extracted from TD1.0 and Δ PS1234 Δ 50k (non-electroporated cells), respectively. As you mentioned, the 4-kb band shows the native plasmid present in *H. bluephagensis*. We have modified the figure to make it clear (See Fig. 8).

Q13: Authors have mentioned in Line 312-increase in surface hydrophobicity leads to increased flocculation. But in this paper, authors have mentioned decrease in hydrophobicity of the muted cells (TEM images). This seems contradictory.

Response: Thanks for your comment. We have included an additional experiment to test the cell surface hydrophobicity. Δ PS124 had increased cell surface hydrophobicity compared with the wild type. The data was included as Fig. 5 and described in Line 208-215. The results of TEM images was explained in Line 205-207. There is no contradiction between these data.

Q14: What is the novelty that authors claim in Line 337?

Response: Thanks for your comment. EPSs normally act as auto agglutinins mediating self-flocculation. In our study, we found the knockout of 3 EPS clusters in *H. bluephagensis* led to self-flocculation. We have done additional experiments to demonstrate the reason causing self-flocculation. The detailed information can be seen in the response to Q3. The novelty of this study was that the lack of EPSs unshielded the more hydrophobic cell surface resulted from O-antigen deficiency and consequently led to self-flocculation of Δ PS124. We have rewritten the novelty in the modified manuscript in Line 378-380.

Q15: Include protocol for RNA extraction in section 4.9.

Response: Based on your suggestion, the protocol for RNA extraction has been included in Line 510-513 as follows. Briefly, total RNA extraction was performed by using TRIzol reagent (TIANGEN Biotech. (Beijing) Co., Ltd.) according to the manufacturer's instructions. mRNA was purified from total RNA using poly-T oligo-attached magnetic beads.

Q16: Include protocol for conjugation in Line 412.

Response: Thanks for your suggestion. We have included this point in Line 451.

Q17: Section 4.2 needs more work. A diagram is needed to explain

Response: Thanks for your comments. We have included a diagram as Supplementary Fig. 5 to clearly explain the construction of plasmids carrying double sgRNA scaffolds and homologous arms based on pSEVA241.

Supplementary Fig. 5. Schematic diagram for the construction of plasmids carrying double sgRNA scaffolds and homologous arms based on pSEVA241.

a. What regions of plasmid was amplified using the primers?

Response: The backbone of pSEVA341 was amplified using primer pair pSEVA241F/ pSEVA241R. The position of the primers has been shown in the diagram.

b. What was the arrangement of the CRISPR array, cas9, in the plasmid?

Response: In this study, a dual-plasmid CRISPR/Cas system was employed for genome editing. Plasmid pSEVA241 was used for carrying and expressing CRISPR arrays as shown in Supplementary Fig. 5. Plasmid pQ08 was used for Cas9 expression (see reference “CRISPR/Cas9 editing genome of extremophile *Halomonas* spp.”). The detailed information can be seen in section 4.4.

c. What are the promoters used?

Response: J23119 promoter was used to express CRISPR arrays. The expression SpCas9 was driven by its native promoter.

d. There should be two different sgRNA's in the double cut experiments. How were

these 2 sgRNA's located in relation to each other?

Response: As you mentioned, two different sgRNAs were used in the double cut experiments. They were arranged next to each other in the same direction in pSEVA241 as shown in Supplementary Fig. 5.

e. What are the constituents of the CRSIPR array?

Response: Each CRISPR array consists of J23119 promoter, gRNA spacer, gRNA scaffold and terminator as shown in Supplementary Fig. 5.

Q18: reference 13, line 391 have no reference to Vector F, and vector R. Please correct the reference or include more details in this manuscript.

Response: The comment raised by the reviewer might be caused by our improper names of primer pair (Vector F/Vector R) which are primers instead of vectors. In order to make it clear we have renamed the two primers as pSEVA241F and pSEVA241R in Supplementary Table 2.

Q19: Include information on how were the sgRNA sequences designed? What is the PAM sequence that is specific to this bacterium?

Response: Thanks for your suggestion. In this study, we used *Streptococcus pyogenes* Cas9 (spCas9) for genome editing and the PAM sequence of spCas9 system is NGG. The sgRNA sequences for the individual target genes were designed based on the principle spCas9. The design of sgRNA has been included in Line 428-430 as follows. For the design of sgRNA spacer, a 20 bp of protospacer with ~ 50% GC content before NGG (PAM) was manually chosen using Snapgene. For the sake of clarity, a diagram including the design of sgRNA sequences has been added as Supplementary Fig. 5.

Q20: Line 201, is it figure 2E or Figure 3? Flocculation is depicted in Figure 3 supplementary.

Response: Thanks for your correction. The figure has been renumbered as Supplementary Fig. 2, which has been correctly cited in the modified manuscript in Line 192.

Reviewer #2 (Remarks to the Author):

In the manuscript entitled “Deficiency of exopolysaccharides makes *Halomonas bluephagenesis* self-flocculating and electrotransformation feasible” the authors present the results of a study in which they adapted a CRISPR-Cas9 strategy to delete large sections of the *H. bluephagenesis* genome. Focus was placed specifically on removing genetic sections associated with the production of exopolysaccharides and the formation of flagella. These regions were selected because of the significant carbon and energetic demands associated with these compounds and organelles. The authors then went on to investigate the effects of these deletions on a range of characteristics, most notably flocculation and electroporation. The work expands from many previous studies from the authors, and the manuscript presents some valuable

interesting findings on a non-traditional bioproduction chassis. The experiments are generally well designed (although some details are missing) however much of the analysis is quite speculative.

Response: Great thanks for your positive comment and constructive suggestions. We have carefully addressed all the comments one by one and further improved this manuscript. Especially, we have performed additional experiments and demonstrated that the increase of cell surface hydrophobicity resulted from the simultaneous deficiency of EPSs and O-antigen is the reason that led to self-flocculation of Δ PS124. We hope the modified manuscript would meet your requirements.

You can find below a list of the main concerns with the study.

1) There are numerous speculative statements throughout the manuscript, which in some cases weaken the quality of the analysis. Some examples include:

Response: Thank you for your helpful comments. We have carefully rewritten the speculative statements of result analysis throughout the manuscript to improve the quality of our analysis.

- lines 147-148: rephrase “because they might possibly be involved” as it is vague and seems unsubstantiated.

Response: We rewrote the phrase as “to interrupt extracellular polysaccharide synthesis.” in Line 147-148.

- lines 173-175: It is not clear that ionic strength is a key factor controlling flocculation. There seems to be a correlation but the experiments performed cannot validate the claim made. Also, the link to DLVO theory should be expanded on in the Discussion.

Response: Thanks for your comment. We have deleted the claim mentioned by you. In the modified manuscript, we have explained how ionic strength influences flocculation based on DLVO theory in Line 325-332 in the discussion section as follows. NaCl concentration influenced the self-flocculation property of *H. bluephagenesis* Δ PS124. This can be explained by the Derjaguin, Landau, Verwey, and Overbeek (DLVO) theory, which is a classical theory of colloidal stability. Bacteria usually exhibit a negative cell surface charge, which attracts the surrounding cations to form a double layer according to the DLVO theory. The size of the double layer decreases with the increase of ionic strength. When the double layer decreases, the repulsion between cells also decreases, thus accelerating cell aggregation.

- lines 183-187: Again, the claim (that EPSs are the most critical factor in self-flocculation) is speculative and, while the evidence might be consistent with the claim, it does not prove it and certainly cannot lead to such a strong statement. This argument is also somewhat discredited in lines 202-204, in which the authors speculate that the process is more complex than expected.

Response: Thanks for your comments. Based on the suggestions from the two

reviewers, we have performed additional experiments to clarify the reason that led to self-flocculation of *H. bluephagenesis*. Our results showed that the lack of EPSs unshielded the more hydrophobic cell surface resulted from O-antigen deficiency and consequently led to self-flocculation of Δ PS124 (Fig. 5 and Fig. 6). We also performed the simultaneous deletion of *fhaB*, *hppD* and *hyp* in Δ PS124, which did not affect the self-flocculation (Supplementary Fig. 2b). Based on the results, we have rewritten the sentence as “The result indicated that the loss of EPSs exposed the hydrophobic cell surface of Δ PS124, which might be closely related with its self-flocculation” in Line 203-205.

- lines 199-200: Speculative statement.

Response: We have deleted this speculative statement in the modified manuscript.

- lines 209-210: change to “This suggests that the absence of EPSs may improve the oxygen permeability of the cell membrane.”

Response: We have changed to “This suggests that the absence of EPSs may improve the oxygen permeability of cell membrane.” in Line 197-198.

- lines 240-243: Could this be substantiated with experimental evidence?

Response: DNA electrotransformation into the EPSs-deleted mutants is feasible. Thus, our speculation is substantiated with the experimental evidence.

- lines 319-323: Some speculation and a shortcut to the last statement.

Response: Based on your comment, we have deleted this sentence in the modified manuscript.

- lines: 349-359: Some speculation and the last sentence could be considered subjective.

Response: Thanks for your comment. We have deleted these subjective sentences in the modified manuscript.

2) lines 61-64: “Minimizing the genome size via deletion of nonessential genes can reduce” and “Genome reduction can generate high-performing”. Caution must be taken here; genome reduction does not systematically lead to such advantages. This is an important distinction and should be addressed in other parts of the manuscript as well (for example in the Discussion lines 276-280).

Response: Thank you for your comments. We have revised the sentences based on your suggestion in Line 63-64 and 306-307 to make the statements more accurate.

3) How extensive was the transcriptomic analysis? Much more information is required on how many replicates were used, how the samples were processed, how the differential expression analysis was performed, which platforms were used for analysis, etc. In the Materials and Methods, the authors divert these aspects by pointing to a third party. This is not sufficient.

Response: Thanks for your suggestions. In the modified manuscript, we have included the detailed information about RNA-Seq analysis as follows (Line 513-517). Each group has two replicates. RNA extraction, transcriptome sequencing, data analysis and differential expression analysis were performed by Novogene Co., Ltd. (Beijing, China)⁴⁹. Briefly, total RNA extraction was performed by using TRIzol reagent (TIANGEN Biotech. (Beijing) Co., Ltd.) according to the manufacturer's instructions. mRNA was purified from total RNA using poly-T oligo-attached magnetic beads. Sequencing was performed on an Illumina Novaseq platform and 2.5 Gb data was generated for each sample. A P value of 0.05 and $|\log_2(\text{fold change})| \geq 1$ were set as the thresholds for significantly differential expression. Differential expression analysis between TD1.0 and Δ PS124 was performed using the DESeq R package (1.18.0).

4) The authors draw a direct relationship between a reduction in OD of the mutants compared to the TD1.0 strain (lines 223-230) and self-flocculation. However, there are no consideration that the smoother cell surface observed through microscopy (Figure 4) may lead to a reduction in light scattering, and hence a reduction in OD₆₀₀ for a given cell number. This would also be consistent with the fact that similar dry cell weights were observed for all three strains even if OD differed.

Additional experiments investigating the relationship between OD and cell numbers for the mutants and TD1.0 would be required to confirm this and validate or reject the claim made by the authors.

Response: Thanks for your comments. In our experiment, we found that OD₆₀₀ of the mutants decreased along with standing time during the measurement process because the mutants rapidly flocculated. The characteristic of self-flocculation might be the main reason causing the OD discrepancy of mutants. As you mentioned, the change of cell surface might also influence the OD₆₀₀ of mutants. Thus, we have included that cell surface change might be another factor causing the difference on OD₆₀₀ in Line 254-255.

Based on your suggestion, we used the gradient dilution coating method to test the relationship between OD₆₀₀ and cell numbers of TD1.0 and Δ PS124 cultivated in LB60 medium. The result was shown in the table below. TD1.0 and Δ PS124 reached a similar OD₆₀₀ (1.29 vs 1.19) at 5 h of cultivation. TD1.0 has a cfu of $(5.31 \pm 0.96) \times 10^8$ per ml, whereas Δ PS124 has only $(3.51 \pm 0.71) \times 10^8$ cfu per ml. Similarly, the cfu deviation of Δ PS124 might be also attributed to its self-flocculation, which disabled a full dispersion of Δ PS124 into single cells in the gradient dilution process.

Table. OD₆₀₀ and cell density of TD1.0 and Δ PS124 at 5 h of cultivation in LB60.

Strain	OD ₆₀₀	cfu/ml	cfu/ml·OD
TD1.0	1.29	$(5.31 \pm 0.96) \times 10^8$	$(4.12 \pm 0.74) \times 10^8$

We then used diphenylamine colorimetric method, which is a more suitable method for self-flocculation cells, to determine the relationship of cell growth with OD₆₀₀ by quantifying DNA. This method has been used for measuring the growth of microbes (*Appl Microbiol Biotechnol* 2013, *Appl Environ Microbiol* 2015, *Appl Environ Microbiol* 2019). As shown in the figure below, the OD₆₀₀ of TD1.0 and Δ PS124 was 7.75 and 6.18 at 24 h of cultivation, respectively. But their OD₅₉₅ per ml culture had a similar value (0.36 vs 0.38). This result showed that TD1.0 and Δ PS124 actually had a similar cell growth and OD₆₀₀ could not accurately reflect the cell growth of Δ PS124 probably due to its self-flocculation. Thus, our claim is consistent with our new result determined by diphenylamine colorimetric method.

Figure. Cell growth shown as OD₅₉₅ per ml culture and OD₆₀₀ of TD1.0 and Δ PS124 at 24 h of cultivation in LB60 medium.

References:

- (1) Zhao Y, Xiang S, Dai X, Yang K. A simplified diphenylamine colorimetric method for growth quantification. *Appl Microbiol Biotechnol* 97(11):5069-5077. (2013).
- (2) Chen, J. et al. Unusual Phosphoenolpyruvate (PEP) Synthetase-like protein crucial to enhancement of polyhydroxyalkanoate accumulation in *Haloferax mediterranei* revealed by dissection of PEP-pyruvate interconversion mechanism. *Appl Environ Microbiol* 85 (2019).
- (3) Hou J, Xiang H, Han J. Propionyl coenzyme A (propionyl-CoA) carboxylase in *Haloferax mediterranei*: indispensability for propionyl-CoA assimilation and impacts on global metabolism. *Appl Environ Microbiol* 81(2):794–804. (2015).

5) It is unclear if statistical analysis was used when comparing values obtained between strains or culture conditions for various experiments. If so, more details should be provided; if not, it should be done. Examples include:

- lines 228-230: “The dry cell weight of the three strains were almost similar (~13 g/L).” This is also put in perspective in the following line where the authors state that TD1.0 had a “slightly higher biomass”. Statistical analysis is required.
- lines 230-231: “Likewise, the PHA production level of TD1.0, ΔPS124 and ΔPS1234 Δ50k was also comparable (~9 g/L)”.
- lines 235-237: mutant strains “could produce a similar amount of PHA by consuming less carbon source”. Again, without proper statistical comparisons of both PHA produced and carbon utilization the claims and analyses made in the manuscript become highly speculative.
- other similar cases can be found throughout the manuscript.

Response: Thanks for your comments. We have double checked this point throughout the manuscript and performed statistical analysis as needed. The method for statistical analysis has been added in the “Materials and method” section.

6) line 124: the location of the gene cluster in the genome should be provided and the acronym CDS should be defined as it is encountered for the first time.

Response: Thank you for your suggestions. We have included the location of the gene cluster in the genome in Fig. 2 and added sequence information in Supplementary data 1. The acronym CDS has been defined as “Coding Sequence” in Line 124.

7) lines 129-130: It should be specified that the rate of mutation reported was specifically for deletions. Unless more information is provided, the authors have not shown whether this rate would also be observed when DNA fragments are inserted.

Response: Thanks for your comment. We have changed “editing efficiency” to “deletion efficiency” throughout the manuscript (Line 123, 123 and 136).

8) lines 249-250: How many cells were used? Should be included.

Response: Thank you for the comment. We have included the number of cells ($1\sim 2\times 10^9$) in Line 280.

9) Table 1 should be reordered for clarity. It is quite confusing as is.

Response: Based on your suggestion, we have reordered Table 1 to make it clear and readable in the modified manuscript.

10) lines 260-265 + Figure 6: Why were only 4 transformants used for these experiments? This is quite low. And does this mean that the authors consider the transformation and stability to be 100%?

A larger number of transformants tested and more replication cycles would have allowed a better statistical assessment and, perhaps, more information on the long-term stability of the plasmid. There is no quantifiable information provided on

these parameters.

Response: Thanks for your suggestion. We have repeated the experiment to evaluate the plasmid stability in more transformants. A total of 46 transformants were randomly selected and cultured overnight in LB60 supplemented with chloramphenicol (25 µg/ml). The plasmids were extracted and analyzed using agarose gel electrophoresis (Supplementary Fig. 4.). The result showed that all the transformants had the plasmid pSEVA341, indicating the stability of pSEVA341 in transformants was 100%.

11) Figure 3A: What was the ionic strength of the medium for the results shown? This should be included in the caption and in the corresponding section of the Discussion. The same should be done for lines 343-344: The conditions used to test flocculation should be specified.

Response: Thanks for your suggestion. The ionic strength of the medium used refers to the concentration of NaCl. The ionic strength of the medium for the results shown in Figure 3A was 1 M NaCl. We have included the information in the legend of Fig. 3A and Line 332 and 461.

12) Were any experiments performed to see if the flocculation effects observed at different ionic strengths were linked to the growth of the cells or only to the given state of the medium (i.e. would cells grown in a medium at a given ionic strength flocculate at a different rate if they were centrifuged and resuspended in a new medium at a different ionic strength?)

Response: Thanks for your comment. The growth of *H. bluephagensis* is inhibited below 0.6 M NaCl and it cannot grow below 0.4 M NaCl. We have tested the self-flocculation of Δ PS124 cultured for 24 h in LB medium containing a wide range of NaCl concentration (0.4 to 1.5 M). Although Δ PS124 reached different cell densities at different NaCl concentration, it could flocculate. The optimum NaCl concentration for *H. bluephagensis* growth is 1.0 M. The fermentation time of *H. bluephagensis* normally lasts for 48 h. From the prospect of industrial application of self-flocculation of Δ PS124, we only tested the flocculation effects of different ionic strengths of Δ PS124 cultured for 48 h in MM60 containing 1 M NaCl.

13) lines 329-330: Individual deletion of the three genes did not affect self-flocculation, but the deletion of two or all three at once may have (which would highlight potential interactions).

Response: Thanks for your comment. We have simultaneously deleted the 3 genes in Δ PS124 and found the deletion did not affect the self-flocculation (Supplementary Fig. 2).

14) line 341: Details of the NPN method should be included.

Response: Thanks for your comment. We have included the details of NPN method in the supplementary materials (Membrane permeability assay).

15) line 410: How was Cas9 transformed in the cells? Was conjugation used? Details should be included.

Response: Thanks for your suggestion. As you mentioned, the Cas9 expression plasmid pQ08 was transformed in the cells by conjugation method. We have included this information to make it clear in Line 451.

16) lines 419-420: include the range of NaCl concentrations used.

Response: According to your comment, we have included the range of NaCl concentrations in Line 461 and the legend of Fig. 3.

17) lines 428-429: Is there a reference for this equation?

Response: Yes, we have included the reference for this equation in Line 470 based on your comment.

Ling, C. *et al.* Engineering self-flocculating *Halomonas campaniensis* for wastewaterless open and continuous fermentation. *Biotechnol Bioeng* 116, 805-815 (2019).

18) lines 430: The title of Section 4.6 is wrong, it should refer to Cultures and/or Cell Dry Weight, not PHA production.

Response: Thanks for your comment. We have changed the title of Section 4.6 to “Shake-flask culture of *H. bluephagenesis*” in Line 471.

19) lines 465-466: General details of GC method used should be included.

Response: Thanks for your comment. We have included the general details of GC method in Line 527-531 as follows. The temperature of flame-ionization detector and injection port were set at 220°C and 200°C, respectively. The temperature of the column was set at 80°C and was maintained for 1.5 min. Then, it rose to 140°C at 30°C/min and further to 220°C at 40°C/min, and was maintained at 220°C for 0.5 min.

20) lines 486-487: Were sequencing and PCR used?

Response: Thanks for your comment. In our experiment, the plasmid electroporated into Δ PS1234 Δ 50k was extracted with Plasmid Extraction Kit and verified by agarose gel electrophoresis. The detailed information has been included in Line 556 and 559 in the modified manuscript.

21) The terms PHB, PHA and PHBV seem to be used interchangeably throughout the manuscript. This should be corrected to the proper molecule produced in the study.

Response: Thanks for your comment. PHB and PHBV are two most common types of PHA. *H. bluephagenesis* used in this study produces PHB. Based on your comment, we have used PHB to indicate the PHA produced by *H. bluephagenesis* throughout the manuscript.

22) The following points should also be addressed.

- line 24: the claim that *H. bluephagenesis* is “one of the most promising cell factories” is very subjective and unfounded. This should be toned down and rephrased here and

at other locations in the manuscript. The same should be done at line 52 where “Extensive research” should be replaced by “Previous work” or “Previous studies have focused on”.

Response: As your suggested, we have rephrased it as “a non-traditional bioproduction chassis” in Line 26 and 411. “Extensive research” has been changed to “Previous work” in Line 54.

- lines 28-29: “including those encoding flagella or exopolysaccharides (EPSs), were”

Response: We have changed to “including those encoding flagella, exopolysaccharides (EPSs) and O-antigen, were” in Line 30-31.

- line 38: “serves as an improved version”

Response: We have changed to “serves as an improved version” in Line 40.

- lines 59-60: Reference(s) needed.

Response: Three references have been included in Line 62.

References:

(1) Jiang, X.R., Yan, X., Yu, L.P., Liu, X.Y. & Chen, G.Q. Hyperproduction of 3-hydroxypropionate by *Halomonas bluephagenesis*. Nat Commun 12, 1513 (2021).

(2) Lin, Y.N. et al. Engineering *Halomonas bluephagenesis* as a chassis for bioproduction from starch. Metab Eng 64, 134-145 (2021).

(3) Ma, H. et al. Rational flux-tuning of *Halomonas bluephagenesis* for co-production of bioplastic PHB and ectoine. Nat Commun 11 (2020).

- line 69: “tools”

Response: We have changed “tool” to “tools” in Line 69.

- line 76: “their origin”

Response: We have changed “its origin” to “their origin” in Line 77.

- lines 113-114: avoid the repetition of “editing”

Response: Thanks for your comment. We have changed the second “editing” to “deletion” in Line 114.

- line 117: The use of “To our knowledge,” is odd since at least one author was also an author on the article referenced.

Response: Thanks for your comment. We have deleted “To our knowledge,” in Line 117.

- lines 118, 119, 122, 124, 177, 404, 406, 455, 470, 479, 480: some characters were replaced by empty checkboxes.

Response: Thanks for your correction. We have double checked the whole manuscript and corrected this error.

- line 164: Change the term “equilibrium”. This is not a situation that leads to an equilibrium. It can lead to a stable state, but not equilibrium.

Response: We have changed “an equilibrium” to “stabilization” in Line 164.

- line 173: “and lost the ability of self-flocculation below 0.2 M NaCl” as it is redundant with lines 168-170.

Response: We have deleted “and lost the ability of self-flocculation below 0.2 M NaCl” in the new manuscript.

- throughout the manuscript: commas designating thousands in numbers should not be followed by a space (e.g. line 146: 16, 861 bp should be 16,861 bp).

Response: Thanks for your corrections. The space after commas designating thousands in numbers has been deleted in Line 146, 147, 151, 242.

- lines 198-199: Reference(s) needed.

Response: Thanks for your comment. We have included two references in Line 190.

(1) Schwibbert, K. et al. A blueprint of ectoine metabolism from the genome of the industrial producer *Halomonas elongata* DSM 2581 T. *Environ Microbiol* **13**, 1973-1994 (2011).

(2) Sarasa, S.B. et al. A Brief Review on the Non-protein Amino Acid, Gamma-amino Butyric Acid (GABA): Its Production and Role in Microbes. *Curr Microbiol* **77**, 534-544 (2020).

- line 205: remove the comma.

Response: We have removed comma in Line 194.

- line 223: specify if the OD600 was final or at stationary phase.

Response: We have made it clear by adding “at stationary phase” in Line 253.

- line 308: change “investment” to “capital and operating” costs.

Response: We have changed “investment costs” to “capital and operating costs” in Line 339.

- line 309: in some processes, the presence of inhibitors or by-products in the medium may limit or prohibit the reuse of medium/supernatant.

Response: Yes, we completely agreed with your opinion. In the case of *Halomonas campaniensis*, the wastewaterless process was conducted for four runs without generating wastewater (see reference “Engineering self-flocculating *Halomonas campaniensis* for wastewaterless open and continuous fermentation”).

- line 313: remove comma.

Response: This sentence has been removed in the modified manuscript.

- line 379: “are shown”.

Response: We have revised “were shown” to “are shown” in Line 416.

- lines 404-405: how was the moss blown up?

Response: We have revised it as “the moss was scraped up” in Line 444.

Reviewer #3 (Remarks to the Author):

This manuscript reports on the development of a strategy for an optimized CRISPR/Cas9 genome editing of a PHB-producing *H. bluephagenesis* strain. Specifically, the authors reported on the use of a dual-sgRNA to delete the redundant synthesis pathways of flagella and EPSs allowing the obtainment of a mutant with enhanced self-flocculation and less carbon and energy requirement for metabolic processes. The mutant was able to accumulate PHB at a percentage similar to WT strain and could be transformed by electroporation.

Overall, the paper is well written, the methodological approach is suitable and the discussion of the obtained results is appropriate. The findings of this work are very interesting considering the importance of next generation industrial biotechnology (NGIB) which aims at developing novel and sustainable approaches, through the selection and engineering of appropriate microbial strains, for the obtainment of chemicals/products overcoming economic and technological challenges of current industrial biotechnology.

Response: Great thanks for your positive feedback and constructive comments that have substantially improved our manuscript.

I found the paper suitable for publication. I have however some specific comments:

1. Pg. 2, line 23 and Pg. 3, line 50: “bacterium” instead of “bacteria”

Response: Thanks for the correction. We have replaced “bacteria” with “bacterium” in lines 25 and 51.

2. Pg. 4, lines 116, 118 and throughout the manuscript: please review the character used (example in the word “efficiency” there is a “square” instead of “ffi”)

Response: Great thanks for your correction. We have double checked the character throughout the whole manuscript and made corrections.

Reviewers' comments:

Reviewer #2 (Remarks to the Author):

Following the review process of the initial submission, the authors have performed and included a significant amount of experimental work and improved their analysis and discussion. All this helps support their claims and the manuscript is much improved by it.

The following changes should be made:

1) The method used to measure the cell surface hydrophobicity measurements reported at lines 211 to 214 should be specified in those sentences. A percentage alone is not clear to the reader. Perhaps modifying the first sentence to include (46.1% as measured by [...]) would be sufficient.

2) The results mentioned in the sentence at lines 223-225 on the complementation of Part 1 are not shown in Figure 6b.

3) I have an issue with the transcriptomic analysis. The authors have now specified that the analysis was done on duplicate samples. Although transcriptomic studies have been published on duplicates, these are inherently statistically flawed. There have been demonstrations that analysis on duplicates often leads to misinterpretations and unreliable comparisons of DEGs. I would remove the results and discussion related to transcriptomics.

Even without this analysis, the manuscript retains value, is still of interest and the conclusions presented are still supported by the rest of the experimental work. The work is still publishable without these results.

4) In section 4.11:

Add all the dimensions of the GC column (ID), carrier gas and gas flow rates. Was a split ratio used?

5) The authors have greatly improved the statistical analysis of the results they presented. However, some statements linked to the comparison of values between treatments have not been modified accordingly and remain vague. This applies in particular to sentences in lines 258-261 and 265-267.

And I would recommend the following:

1) I would recommend modifying the title to:

"Deficiency of exopolysaccharides and O-antigen makes *Halomonas bluephagenesis* self-flocculating and amenable to electrotransformation"

2) Line 336: clarify

3) Line 379: " Δ PS124

Also remove: "which is caused by the enhanced hydrophobic cell surface due to the lack of both EPSs and O-antigen."

Point-to-Point Response to Reviewers' Comments

(Manuscript ID# COMMSBIO-21-1273A)

Editor's comments

Your manuscript entitled "Deficiency of exopolysaccharides and O-antigen makes *Halomonas bluephagenesis* self-flocculating and electrotransformation feasible" has now been seen by 1 referees. You will see from their comments below that while they find your work of considerable interest, some important points are raised. We are interested in the possibility of publishing your study in *Communications Biology*, but would like to consider your response to these concerns in the form of a revised manuscript before we make a final decision on publication.

We therefore invite you to revise and resubmit your manuscript, taking into account the points raised. In particular, we ask that you address their comment regarding the number of replicates used for the transcriptomics, and include a discussion of this limitation in the manuscript.

Response: We thank you for the positive feedback. We have carefully addressed each comment raised by the reviewer. We hope the revised manuscript will meet the standards required by *Communications Biology*, you and the reviewer.

Reviewers' comments:

Reviewer #2 (Remarks to the Author):

Following the review process of the initial submission, the authors have performed and included a significant amount of experimental work and improved their analysis and discussion. All this helps support their claims and the manuscript is much improved by it.

Response: Great thanks for your positive comments and helpful recommendations.

The following changes should be made:

1. The method used to measure the cell surface hydrophobicity measurements reported at lines 211 to 214 should be specified in those sentences. A percentage alone is not clear to the reader. Perhaps modifying the first sentence to include (46.1% as measured by [...]) would be sufficient.

Response: Thanks for your recommendation. Based on your suggestion, the first sentence has been modified by including "(46.1% as measured by a modified phase partitioning assay)" (Line 211-212).

2. The results mentioned in the sentence at lines 223-225 on the complementation of Part 1 are not shown in Figure 6b.

Response: Thanks for your comment. As you suggested, the result about complementation of Part1 on self-flocculation has been included in Supplementary Fig. 2c (Line 225 and Supplementary Information).

3. I have an issue with the transcriptomic analysis. The authors have now specified that the analysis was done on duplicate samples. Although transcriptomic studies have been published on duplicates, these are inherently statistically flawed. There have been demonstrations that analysis on duplicates often leads to misinterpretations and unreliable comparisons of DEGs. I would remove the results and discussion related to transcriptomics.

Even without this analysis, the manuscript retains value, is still of interest and the conclusions presented are still supported by the rest of the experimental work. The work is still publishable without these results.

Response: Thanks for your suggestion and positive comments. We agree with the reviewer that the number of replicates for transcriptomic studies is one of the crucial factors influencing DEG analysis (see reference “A survey of best practices for RNA-seq data analysis” published in Genome Biology). Two replicates may reduce the capacity for detecting statistically significant differences in gene expression between experimental groups.

In our case, two replicates are used for RNA-seq experiment and the sequencing depth for each sample is ~7 millions of reads. The data from the two samples have good repeatability and can meet our requirements for screening the most significant differential genes probably involved in self-flocculating. Although knockout of the three significantly up-regulated genes did not affect self-flocculation, the data may have some reference values for readers. Therefore, it is better not to delete this data from the MS.

4. In section 4.11:

Add all the dimensions of the GC column (ID), carrier gas and gas flow rates. Was a split ratio used?

Response: Based on the comment, we have added the detailed information as “(30 m × 0.32 mm × 0.25 μm film thickness, part number 123-3232, Agilent, USA). High purity N₂ was used as carrier gas and the column head pressure of N₂ carrier gas was set to 10 psi. A total of 1 μL of sample was injected with splitless mode.” (Line 527-530).

5. The authors have greatly improved the statistical analysis of the results they presented. However, some statements linked to the comparison of values between treatments have not been modified accordingly and remain vague. This applies in particular to sentences in lines 258-261 and 265-267.

Response: Thanks for your comments. We have deleted the sentence “The slightly

higher biomass of TD1.0 might be due to its EPSs synthesis” from the manuscript (Line 261). Also, we have deleted “or Δ PS1234 Δ 50k” from “This result indicated that mutant strain Δ PS124 or Δ PS1234 Δ 50k could produce a similar amount of PHB by consuming less carbon source compared to TD1.0.” (Line 267).

And I would recommend the following:

6. I would recommend modifying the title to:

“Deficiency of exopolysaccharides and O-antigen makes *Halomonas bluephagenesis* self-flocculating and amenable to electrotransformation”

Response: Thanks for your recommendation. We have changed the title to “Deficiency of exopolysaccharides and O-antigen makes *Halomonas bluephagenesis* self-flocculating and amenable to electrotransformation”.

7. Line 336: clarify

Response: Thanks for the correction. We have corrected the word (Line 357).

8. Line 379: " Δ PS124

Also remove: “which is caused by the enhanced hydrophobic cell surface due to the lack of both EPSs and O-antigen.”

Response: Thanks for the correction. We have changed “PS124” to “ Δ PS124” and removed “which is caused by the enhanced hydrophobic cell surface due to the lack of both EPSs and O-antigen” from this sentence (Line 380).